# Convergent evolution in Afrotheria and non-afrotherians demonstrates high evolvability of the mammalian inner ear

Nicole D. S. Grunstra [1,2,3] ✉, Fabian Hollinetz[1], Guillermo Bravo Morante[1],
Frank E. Zachos [1,3,4,5], Cathrin Pfaff [6], Viola Winkler [7],
Philipp Mitteroecker[1,2,8,10] & Anne Le Maître [1,2,8,9,10] ✉

Evolutionary convergence in distantly related species is among the most convincing evidence of adaptive evolution. The mammalian ear, responsible for balance and hearing, is not only characterised by its spectacular evolutionary incorporation of several bones of the jaw, it also varies considerably in shape across modern mammals. Using a multivariate approach, we show that in Afrotheria, a monophyletic clade with morphologically and ecologically highly disparate species, inner ear shape has evolved similar adaptations as in non-afrotherian mammals. We identify four eco-morphological trait combinations that underlie this convergence. The high evolvability of the mammalian ear is surprising: Nowhere else in the skeleton are different functional units so close together; it includes the smallest bones of the skeleton, encapsulated within the densest bone. We suggest that this evolvability is a direct consequence of the increased genetic and developmental complexity of the mammalian ear compared to other vertebrates.

Some of the most spectacular macroevolutionary changes in the vertebrate ear – responsible for balance, posture, gaze stabilisation, and hearing – occurred in mammals. The transformation of jaw bones into the middle ear ossicles, which evolved at least three times independently in mammals, greatly increased the bandwidth of hearing[1,2]. Compared to other tetrapods, mammals possess two extra middle ear ossicles. The shape of the mammalian inner ear also deviates from that of other tetrapods, notably in the length of the cochlea and the evolutionary novelty of the organ of Corti with its electromotile outer hair cells[3,4]. The length and curvature of the cochlear canal is much more diverse in mammaliaforms compared to other groups[5–9]. Extensive cochlear coiling even evolved multiple times independently in mammals[6,9]. Mammals are also the only group to have co-opted the

angular bone of the jaw into a bone specialised in supporting the eardrum, the ectotympanic bone[10]. Not only is the mammalian ear system thus more heterogeneous and morphologically complex, it is also genetically and developmentally more complex than in other tetrapods[11,12]. For example, only in mammals do ear components derive from both the first and second pharyngeal arches (in all other tetrapods only the second arch is involved) and thus involve cell lineages from additional origins and more diverse gene expression patterns[12,13]. We previously proposed that this increased heterogeneity and developmental complexity of the mammalian ear bestows it a higher developmental modularity and thus an increased evolvability, i.e., an enhanced capacity for adaptive evolution[14–16]. This increase in evolvability may have facilitated the evolution of different hearing

[1]Department of Evolutionary Biology, University of Vienna, Vienna, Austria. [2]Human Evolution and Archaeological Sciences (HEAS), University of Vienna, Vienna, Austria. [3]Mammal Collection, Natural History Museum Vienna, Vienna, Austria. [4]Department of Genetics, University of the Free State, Bloemfontein, South Africa. [5]Research Institute for the Environment and Livelihoods, Charles Darwin University, Casuarina, NT, Australia. [6]Department of Palaeontology, University of Vienna, Vienna, Austria. [7]Central Research Laboratories, Natural History Museum Vienna, Vienna, Austria. [8]Konrad Lorenz Institute for Evolution and Cognition Research, Klosterneuburg, Austria. [9]Laboratoire Paléontologie Evolution Paléoécosystèmes Paléoprimatologie (PALEVOPRIM) – UMR 7262 CNRS INEE, Université de Poitiers, Poitiers, France. [10]These authors contributed equally: Philipp Mitteroecker, Anne Le Maître. ✉e-mail: nicole.grunstra@univie.ac.at; anne.le-maitre@univie.ac.at

adaptations in the auditory system of the middle and inner ear, as well as locomotor adaptations in the vestibular system through its integration with the auditory system. Ultimately, this enhanced evolvability may have contributed to the evolution of the spectacular disparity in mammalian body plans and ecological niches[17] (see also below).

The relative size and shape of the bony labyrinth (the bony wall surrounding the inner ear) differs considerably across mammalian lineages, which is assumed to reflect evolutionary adaptations to different locomotor behaviours, ecology, and hearing capacities in mammals e.g.,[7,18–22], and posture[23,24] (but see refs. [25,26]). For example, the semicircular canals, important for balance and posture control, are strongly reduced in size in cetaceans, which has been linked to their aquatic lifestyle[6,27]. In fast and manoeuvrable terrestrial taxa, the semicircular canals are large relative to body mass and particularly thin[28]. The length and coiling of the cochlea of the inner ear (responsible for the detection and transmission of sound) is largest in mammals that rely heavily on hearing for prey or predator detection and environmental navigation, such as in bats and subterranean rodents[6,29]. An extended secondary bony lamina of the cochlea is associated with ultrasonic hearing in cetaceans[30–32]. Apart from these strong functional signals, bony labyrinth morphology is also indicative of phylogenetic relatedness e.g.,[6,21,22,31,33–35]. The bony labyrinth has therefore been intensively studied so as to reconstruct both environmental adaptations and the phylogenetic position of fossil mammals e.g.,[18,33,35–41].

By comparison, the bony labyrinth of Afrotheria – one of the four main clades of placental mammals – has been understudied (but see refs. [33,36–38]). Relative to the other main clades (Xenarthra, Archontoglires, and Laurasiatheria), Afrotheria show a high disparity to diversity ratio, i.e., they exhibit a striking degree of ecological and morphological disparity while comprising only few species (no more than 90). The crown group Afrotheria evolved on the African continent (including Madagascar), from which the clade derives its name, between ~90–70 Ma in the late Cretaceous[42–44]. Modern Afrotheria includes six higher taxa traditionally classified as orders that bear little resemblance to each other: Proboscidea (elephants), Hyracoidea (hyraxes), Sirenia (sea cows), Tubulidentata (the aardvark), Afrosoricida (golden moles, tenrecs, and otter shrews), and Macroscelidea (elephant shrews or sengis). They include fully aquatic (sea cows), semi-aquatic (otter shrew and some tenrecs), terrestrial (e.g., elephants, aardvark), arboreal (tree hyrax) and subterranean taxa (golden moles). Extant afrotherians also vary in body mass by more than five orders of magnitude (~40 g for an elephant shrew up to >4000 kg for an elephant). Many ecological and morphological similarities with non-afrotherian placentals exist, and until molecular phylogenetics resolved them as a separate clade, the different afrotherian subgroups were scattered throughout the placental mammalian phylogeny[45]. Notably, the phylogenetic position of golden moles and tenrecs in Afrotheria long evaded detection due to their strong morphological similarity to 'true' moles, hedgehogs and shrews[46–48]. Also fossil afrotherians are known for their morphological similarities to distant relatives, such as the gazelle-like hyrax *Antilohyrax*, the rhino-like afrotherian *Arsinoitherium* that is related to elephants, and *Moeritherium*, a hippo-like proboscidean that also shows some similarities to pigs and tapirs[49–51]. Convergent evolution even appears to have taken place *within* Afrotheria: the extinct Malagasy *Plesiorycteropus* bears a strong resemblance to the living aardvark (*Orycteropus*) as well as to living tenrecs[33,52].

The discordance between molecular and morphological similarities among Afrotheria and the many morphological homoplasies shared with other placental mammals attest to the high degree of evolutionary convergence in the external and internal morphology of Afrotheria. Also the bony labyrinth is morphologically highly disparate across Afrotheria[6,33], and some evolutionary convergences have been described, such as adaptations to low-frequency hearing in both Proboscidea and the fossil clade Embrithopoda[37] as well as convergences related to an aquatic lifestyle in Sirenia and the non-afrotherian clade Cetacea[36].

Given the hypothesized high evolvability of the mammalian ear relative to other vertebrates[17] and the striking disparity of Afrotheria, here we study if the bony labyrinth evolved similar functional adaptations in afrotherian and non-afrotherian mammals. To this end, we compare the bony labyrinth shape of 20 afrotherian taxa (covering 22-25% of extant afrotherian species) to that of 20 other, morphologically and ecologically analogous mammals using 3D geometric morphometric data (Figs. 1a and 2) and 12 contextual variables that measure ecological and behavioural properties of the species. Additionally, we group the 40 species into 11 pairs of afrotherian and non-afrotherian species that are morphologically, ecologically and/or behaviourally analogous (Fig. 1b). We analyse these data by a new multivariate approach. Comparing ecologically analogous species one by one or in terms of simple clusters in shape space can be ineffective, because convergent evolution does not necessarily yield perfectly similar, statistically indistinguishable traits in convergent taxa ('incomplete convergence' sensu[53,54]), and because adaptive and phylogenetic signals may overlap as a result of adaptive diversification. Instead, we first quantify the overall magnitude of morphological convergence within the sample by multivariate summary statistics. After having quantified the *pattern* of convergence, we assess the underlying adaptive *process*[54]. To this end, we identify the bony labyrinth traits that drive the convergence by estimating shape features with maximal association with the contextual variables (presumably adaptive traits). Finally, after removing this trait variation from the data, we investigate if the signal of convergence is lost in order to assess if these traits completely underlie the observed pattern of convergence.

Using this approach, we are able to show that afrotherians resemble their non-afrotherian analogues more closely in overall labyrinth shape than they do each other or non-afrotherian non-analogues. Despite their phylogenetic distance, we found afrotherians and their analogues to share the same associations of bony labyrinth shape with ecology and positional behaviour (posture and locomotion). The entire signal of convergence is captured by four dimensions of bony labyrinth shape variation. These results reflect the dominance of functional adaptation and convergence over phylogenetic history in inner ear shape variation at the level of major placental mammal clades, illustrating the high evolvability of the mammalian ear.

## Results

### Evolutionary convergence of overall labyrinth shape

To assess the overall convergence of bony labyrinth shape, we calculated the Procrustes distance (a measure of overall shape dissimilarity) between all pairs of afrotherians and non-afrotherians. The average Procrustes distance between afrotherians and their non-afrotherian analogues (Fig. 1, Table 1) was 0.245, whereas the average distance between afrotherians and (non-afrotherian) non-analogues was 0.286. In other words, the average shape distance for pairs of non-analogues was 17.3% higher than for pairs of analogues ($p < 0.001$), reflecting convergent evolution in Afrotheria and other mammals. Even the average Procrustes distances among all pairs of Afrotheria (0.266), i.e., between afrotherians and other afrotherians, and among all pairs of non-afrotherians (0.290) exceeded the average distance between Afrotheria-analogue pairs, showing that functional adaptation dominates over phylogenetic history in the average magnitude of overall labyrinth similarity at this high taxonomic level. When statistically correcting for phylogenetic relatedness, the ratio of the average Procrustes distance between pairs of analogues to the average Procrustes distance between all species pairs has been referred to as the Wheatsheaf index[55]. For our data, this index amounts to 1.183 and thus is close to the uncorrected ratio. Similar results are obtained when comparing median instead of mean Procrustes distances.

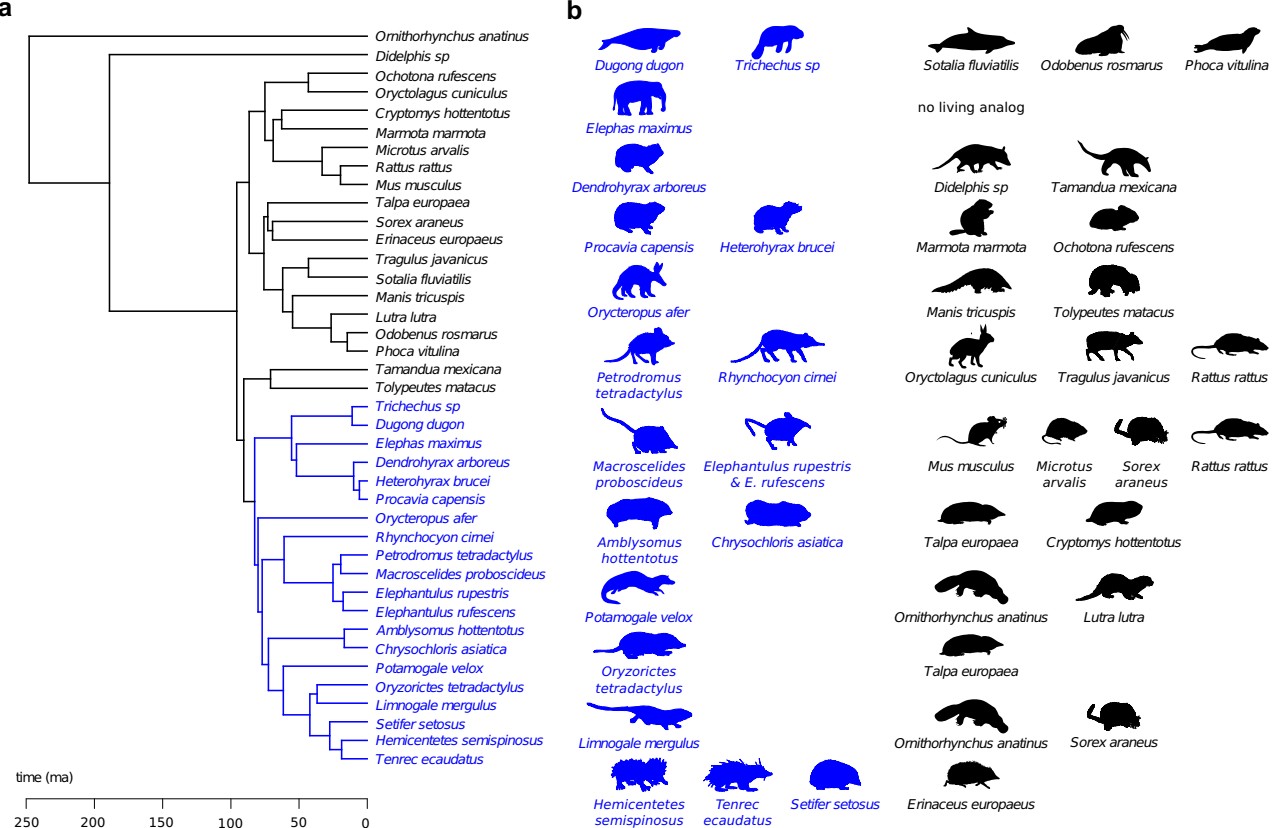

**Fig. 1 | Phylogeny and silhouettes depicting the sample of Afrotheria and their analogues. a** Phylogenetic relationships between Afrotheria (blue) and the other mammals in our sample (black), for which we follow the Atlantogenata hypothesis. **b** Taxa are depicted as their silhouettes, with each row showing the afrotherians (blue) and their ecological and/or morphological analogue(s) (black). The basis of analogy was overall body morphology (e.g., the greater hedgehog tenrec, *Setifer setosus*, and the true hedgehog, *Erinaceus europaeus*), specific morphological traits (e.g., relative hindlimb length as in the larger-bodied elephant shrew, *Rhynchocyon*

*cirnei*, the mouse deer, *Tragulus javanicus*, and rabbit, *Oryctolagus cuniculus*), and/ or locomotion and habitat (e.g., the sea cows and the dolphin). See Table 1 for more details. Silhouettes are by the authors and from www.phylopic.org. Credits go to Kai Caspar for *C. hottentotus* (creativecommons.org/licenses/by/3.0/), to Roberto Díaz Sibaja for *E. europaeus* (creativecommons.org/licenses/by/3.0/), to Sarah Werning for *O. anatinus* (creativecommons.org/licenses/by/3.0/), and to Chris Huh for *S. fluviatilis* (creativecommons.org/licenses/by-sa/3.0/). No changes to these silhouettes were made.

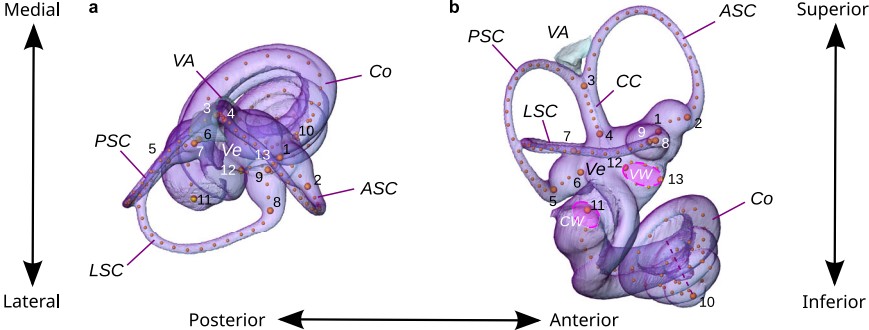

**Fig. 2 | Landmark scheme used to quantify the shape of the bony labyrinth. a** Superior view; **b** lateral view. The bony labyrinth, i.e., the osseous moulding of the inner ear, of a Javan mouse deer (*Tragulus javanicus*). Linear accelerations of the head are detected in the vestibule (Ve), at the base of which the vestibular aqueduct (VA) inserts, and head rotations are detected at the bulged base (ampulla) of each of the three semicircular ducts. The bony counterparts of these ducts are the lateral, anterior, and posterior semicircular canals (LSC, ASC, and PSC), with the

latter two fused to form the common crus (CC). Sounds are detected in the cochlea (Co) by the propagation of sound pressure waves from the vestibular window (VW) to the apex of the cochlea, then back to the cochlear window (CW) through which they dissipate. To quantify bony labyrinth shape, we placed 13 anatomical landmarks (large numbered spheres) and 111 semilandmarks (smaller spheres), mostly on the centrelines of the semicircular canals and the cochlea, on every CT scan.

**Table 1 | Afrotherian and non-afrotherian mammals used in this study, including the basis for analogy**

| Afrotheria (common name) | Non-afrotherian analogues (common name) | Basis for analogy |
|---|---|---|
| *Hemicentetes semispinosus* (streaked tenrec), *Setifer setosus* (greater hedgehog tenrec), *Tenrec ecaudatus* (tailless tenrec) | *Erinaceus europaeus* (European hedgehog) | Overall appearance, body shape and size, foraging behaviour, terrestrial locomotion with climbing ability |
| *Limnogale mergulus* (web-footed tenrec) | *Ornithorhynchus anatinus* (duck-billed platypus), *Sorex araneus* (Eurasian shrew) | Morphology (*S. araneus*), semi-aquatic lifestyle and habitat (*O. anatinus*) |
| *Orizoryctes tetradactylus* (four-toed rice tenrec) | *Talpa europaea* (European mole) | Overall appearance, fossorial locomotion |
| *Potamogale velox* (otter shrew) | *Lutra lutra* (Eurasian otter), *Ornithorhynchus anatinus* (duck-billed platypus) | Semi-aquatic lifestyle, overall appearance (*L. lutra*), foraging behaviour |
| *Amblysomus hottentotus* (Hottentot golden mole), *Chrysochloris asiatica* (Cape golden mole) | *Cryptomys hottentotus* (common mole-rat), *Talpa europaea* (European mole) | Overall morphology, body size, fossorial locomotion, subterranean habitat |
| *Elephantulus rufescens* (East African long-eared elephant shrew), *Elephantulus rupestris* (Western rock elephant shrew), *Macroscelides proboscideus* (round-eared elephant shrew) | *Microtus arvalis* (common vole), *Mus musculus* (house mouse), *Rattus rattus* (black rat), *Sorex araneus* (Eurasian shrew) | Overall body shape and size (except *R. rattus*), fast terrestrial locomotion (although no living 'micro-cursorial' mammals other than elephant shrews exist) |
| *Petrodromus tetradactylus* (four-toed sengi), *Rhynchocyon cirnei* (checkered sengi) | *Oryctolagus cuniculus* (European rabbit), *Rattus rattus* (black rat), *Tragulus javanicus* (Javan mouse deer) | Fast, terrestrial locomotion, relative hindlimb size (*O. cuniculus, T. javanicus*) |
| *Orycteropus afer* (aardvark) | *Manis tricuspis* (tree pangolin), *Tolypeutes matacus* (southern three-banded armadillo) | Fossorial/burrowing locomotion, foraging behaviour |
| *Heterohyrax brucei* (bush hyrax), *Procavia capensis* (rock hyrax) | *Marmota marmota* (alpine marmot), *Ochotona rufescens* (Afghan pika) | Rocky/open/mountainous habitat, terrestrial and climbing locomotion |
| *Dendrohyrax arboreus* (tree hyrax) | *Didelphis* sp. (opossum) *Tamandua mexicana* (northern tamandua) | Arboreality, climbing locomotion |
| *Dugong dugon* (dugong), *Trichechus* sp. (manatee) | *Odobenus rosmarus* (walrus), *Phoca vitulina* (harbour seal), *Sotalia fluviatilis* (tucuxi/grey dolphin) | Aquatic lifestyle, overall body size and shape (*O. rosmarus*), foraging behaviour (*O. rosmarus*) |
| *Elephas maximus* (Asian elephant) | n/a | n/a |

All afrotherian orders are represented. Higher taxonomic groups of the analogues are as follows: Xenarthra (*Tamandua mexicana, Tolypeutes matacus*), Carnivora (*Lutra lutra, Phoca vitulina, Odobenus rosmarus*), Pholidota (*Manis tricuspis*), Cetartiodactyla (*Tragulus javanicus, Sotalia fluviatilis*), Eulipotyphla (*Erinaceus europaeus, Sorex araneus, Talpa europaea*), Rodentia (*Mus musculus, Rattus rattus, Microtus arvalis, Marmota marmota, Cryptomys hottentotus*), Lagomorpha (*Oryctolagus cuniculus, Ochotona rufescens*), Marsupialia (*Didelphis* sp.), and Monotremata (*Ornithorhynchus anatinus*).

## Adaptive signals in labyrinth shape

To identify the features that underlie this convergence in bony labyrinth shape, we performed a two-block partial least squares analysis (2B-PLS) between all 372 Procrustes shape coordinates and 12 contextual variables that capture multiple aspects of ecology and positional behaviour (Table 2).

The first four dimensions (or pairs of latent variables) together accounted for 93.8% of the summed squared covariances between the two blocks of variables (hierarchical permutation tests against a null hypothesis of no association yielded $p$ values of 0.001, 0.030, 0.043, and 0.103, respectively, for the four dimensions). Further dimensions accounted for only 2% or less. PLS 1 (53.8% of the summed squared covariances) contrasted aquatic species from terrestrial species (Fig. 3). Aquatic and semi-aquatic species had smaller and rounder semicircular canals, particularly the posterior canal, which was shifted superiorly relative to the lateral canal. These features were associated with a larger, rounder vestibular window and a broad, flat cochlea with fewer turns. PLS 2 (20.1% of the summed squared covariances) distinguished fossorial and pronograde species from arboreal and scansorial taxa and from species moving in three dimensions in general. Fossorial species, especially fully subterranean ones, had a long common crus, nearly as high as the vertical (anterior and posterior) semicircular canals, and an acute angle between these canals compared to 3D-moving, arboreal, and scansorial species. They also had a more posteriorly projected lateral semicircular canal, as well as a narrower cochlea with more turns.

PLS 3 (13.3% of the summed squared covariances) contrasted agile species that pursue prey on the ground – and some agile aquatic species – with leaping or jumping arboreal species and subterranean, fossorial species (Fig. 4). Along PLS 3, bony labyrinth shape mainly varied in the relative expansion of the semicircular canals: superior for the vertical canals and the common crus, and lateral for the lateral

canal. Taxa with expanded canals also had a cochlea with fewer turns and a loose coiling. PLS 4 (6.6% of the summed squared covariances) was a contrast between slow, pronograde and to some extent large species (mainly driven by the elephant) versus leaping, jumping, and agile species moving in 3D. Along PLS 4, species varied in the anterior-posterior expansion of the anterior part of the bony labyrinth (i.e., the anterior and lateral semicircular canals, vestibular window and cochlea), relative to its posterior part.

We also conducted a phylogenetic 2B-PLS analysis (Supplementary Figs. 3 and 4), which showed that similar associations remain after accounting for phylogeny, though the ecomorphological associations are captured by the four PLS dimensions in slightly different ways.

## Phylogenetic signals in labyrinth shape

Despite these patterns of convergent evolution, signals of phylogenetic history were also visible. Closely related species with relatively recent divergence dates (e.g., the golden moles, the elephant shrews, or the hyraxes) had similar bony labyrinth morphologies and ecologies (Figs. 3F and 4F, Supplementary Figs. 1 and 2). Such phylogenetic patterning at lower taxonomic levels (e.g., within an order) was supported by statistically significant phylogenetic signal in the shape scores of the four PLS dimensions; Blomberg's $K$ was 0.52, 0.7, 0.48, and 0.43, respectively. But closely related species also tended to share a similar environment or lifestyle; the context scores showed $K$-values from 0.38-0.69. These results indicate that adaptive evolution in the inner ear of mammals reflects a combination of phylogenetic divergence (whereby closely related taxa share adaptations to a similar environment) and convergence (whereby distantly related species share adaptations to a similar environment), i.e. incomplete convergence.

Given the importance of habitat and locomotor behaviour to a species' morphological adaptations, the four PLS dimensions

**Table 2 | The 12 contextual variables of body size, ecology, and positional behaviour used in this study**

| Type | Variables $p$ | Measurement levels | Notes |
|---|---|---|---|
| Body size | $p = 1$ (body mass) | Continuous (in g) | |
| Food acquisition | $p = 1$ (pursuit of moving prey) | 1 = never or rarely<br>2 = mixed diet / sometimes<br>3 = often or always | See text for further explanation. |
| Habitat type | $p = 3$ (aquatic; ground-dwelling; arboreal) | 1 = never or rarely<br>2 = sometimes<br>3 = often or always.<br>For aquatic only:<br>1 = never or rarely<br>2 = moves freely on land and in water<br>3 = mainly aquatic, only rests on land<br>4 = strictly aquatic. | Subterranean habitats are captured by a fossorial mode of locomotion (see below). |
| Mode of locomotion | $p = 4$ (scansorial/climbing; cursorial/running; leaping/jumping/ hopping; fossorial/burrowing) | 1 = typical/frequent<br>2 = occasional<br>3 = atypical/rare. | |
| Dimensionality of movement | $p = 1$ (moves three-dimensionally) | 1 = never or rarely<br>2 = sometimes<br>3 = often or always. | Refers to whether animals move in 3D at any given point in time. |
| Agility | $p = 1$ (agility) | 1 = extra slow<br>2 = slow<br>3 = medium slow<br>4 = medium<br>5 = medium fast<br>6 = fast | Capturing both speed and manoeuvrability. Adapted from Spoor et al[28]. |
| Posture | $p = 1$ (orthogrady vs. pronogrady) | 1 = fully orthograde<br>2 = mostly orthograde, sometimes pronograde<br>3 = mixed<br>4 = mostly pronograde, sometimes orthograde<br>5 = fully pronograde | Orthograde = trunk is upright; pronograde = trunk is horizontal. No taxa in this study are fully orthograde. |

With the exception of body mass, all variables were scored on an ordinal scale.

described above estimate the functional aspects of labyrinth shape and – by and large – separate the functional part of shape space from those shape features that carry no or little functional signal. When computed only from these four PLS dimensions, the average shape distance for pairs of non-analogues was 42.0% higher than for pairs of analogues ($p < 0.001$), reflecting strong convergent evolution of these shape features in ecologically, behaviourally, and morphologically similar mammals. By contrast, after removing these four dimensions from the data, the average Procrustes distances of non-analogue pairs and analogue pairs were of similar magnitude (0.197 versus 0.190; $p = 0.13$). This indicates that the pattern of convergence is successfully captured by the four PLS dimensions.

## Stability of the results

We assessed if our results were stable despite the limited sample size in combination with the high morphological and ecological disparity within this sample (where a few species might unduly influence the results). To this end, we performed a leave-one-out cross-validation (also known as jackknife resampling) and a leave-two-out cross-validation. That is, we repeated the analyses for all possible samples where one or two species are left out, respectively. For the comparison of average Procrustes distances, we additionally repeated the analyses when entire analogue pairs are left out, to evaluate whether a single case of strong evolutionary convergence unduly affected the average Procrustes distance between afrotherians and their analogues. All these replicates led to results very similar to those presented here and support the same interpretation of the data (see Supplementary Figs. 5–7).

## Discussion

Convergent adaptations in distantly related species to shared environmental or functional regimes is among the most convincing

evidence of adaptive evolution. Here we have introduced a multivariate approach to study both the pattern and the process of evolutionary convergence and showed that in Afrotheria, a monophyletic clade with morphologically and ecologically highly disparate species, bony labyrinth shape evolved similar adaptations as in non-afrotherian mammals. We identified four dimensions of shape space, i.e., four combinations of bony labyrinth traits, that underlie this evolutionary convergence.

Compared to other species, all strictly aquatic marine mammals (the manatee, dugong, and dolphin) had extremely small semicircular canals, leading to a low sensitivity to head rotations. This feature has been proposed to be an adaptation for moving in all directions under water while having limited flexibility of the neck, which removes the need for stabilisation of the head through the vestibulo-collic reflex[27,56] (but see Ref. 57). All aquatic and semi-aquatic mammals had a cochlea with a limited number of turns (less than two, except for the walrus and the otter), and hence a short cochlear canal, which is linked to a reduced sensitivity to low-frequency sounds in terrestrial mammals[58,59]. Most of them also had a broad basal cochlear turn that is not tightly coiled with the next one, a feature associated with high-frequency hearing in aquatic mammals[60–62]. Arboreal mammals, especially the scansorial taxa, were characterised by a short common crus, and hence reduced mechanical coupling between the two vertical semicircular canals[63]. They also tended to have small semicircular canals, which leads to low sensitivity and might avoid flow disturbance in the ducts in case of vigorous rotations, and thus helps to prevent overreactions to head movements while balancing on, e.g., tree branches[20,57]. Leaping and jumping arboreal taxa had a long cochlear canal with a graded curvature, indicating good low-frequency hearing and sound localisation abilities[58,59,64]. On the contrary, all subterranean taxa have evolved a long common crus allowing for a strong coupling

 

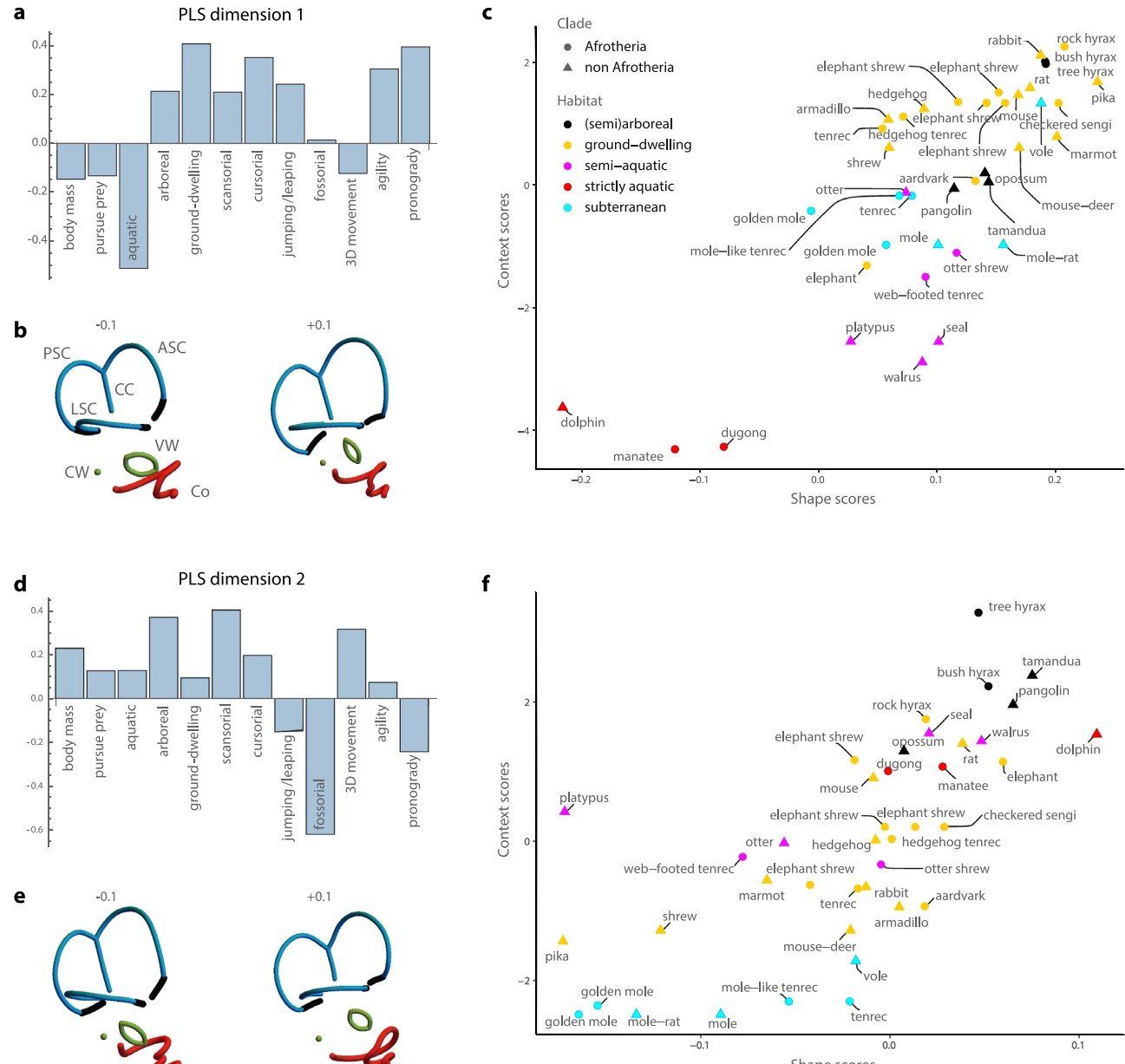

**Fig. 3 | Results for the partial least squares analysis of bony labyrinth shape and the contextual variables (PLS 1 and 2).** First two dimensions of the two-block partial least squares (2B-PLS) analysis between the Procrustes shape coordinates of the bony labyrinth and the 12 contextual variables. **a, b** Loadings for the first PLS dimension, representing the shape pattern and the contextual pattern with highest covariance. LSC, ASC, and PSC correspond to lateral, anterior, and posterior semicircular canals (in blue), Co refers to the cochlea (in red), and VW and CW to the vestibular and cochlear windows (in green). The orientation of the bony labyrinths is the same as in Fig. 2b. **c** Scatterplot of the corresponding contextual and shape scores. The colour code corresponds to five habitat types which were assigned to species as a visual aid only; this grouping was not used for computing the PLS. Note that when a species' common name appears several times, it corresponds to different species. **d–f** Loadings and scores for the second PLS dimension. Source data are provided in the Source Data file.

of the vertical semicircular canals, and hence an efficient detection of vertical head movements[63]. This is likely important for detecting the direction of gravity (vertical acceleration) in environments with few other cues for orientation. Most subterranean taxa had very large semicircular canals, yielding increased sensitivity to head rotations in all directions[20,57,65]. Additionally, subterranean taxa were characterised by a very long cochlea with many turns, leading to a high sensitivity to low-frequency sounds, which attenuate less rapidly in underground environments[6,19,29,66]. Finally, agile species that pursue moving prey have evolved very large semicircular canals, conferring increased sensitivity to head rotations in all directions, which is necessary for fast

movements and good reflexes with respect to posture, balance and eye movements[28,57].

Because the signal of overall convergence (as measured by average Procrustes distances) diminished after removing the aforementioned four dimensions from the data, adaptive evolution in labyrinth shape seems to be largely captured by these four shape patterns. These patterns were estimated as linear combinations with maximal covariance with the contextual variables, but this does not necessarily imply that our contextual variables represent the actual (or all) functional parameters that were under selection. For example, we did not have the actual auditory parameters that describe a species' hearing ability

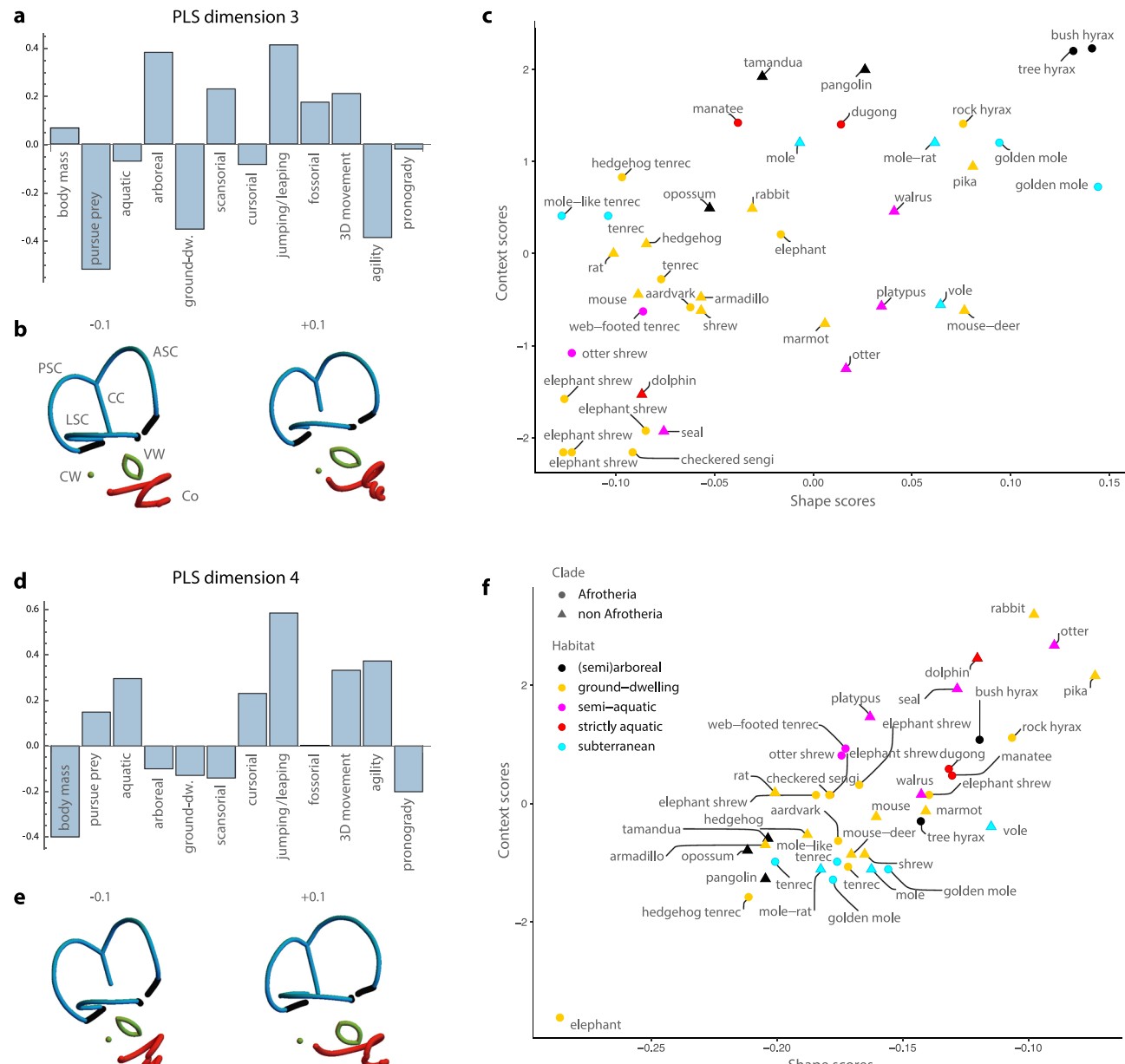

**Fig. 4 | Results for the partial least squares analysis of bony labyrinth shape and the contextual variables (PLS 3 and 4).** Dimensions 3 and 4 of the two-block partial least squares (2B-PLS) analysis between the Procrustes shape coordinates of the bony labyrinth and the 12 contextual variables. **a**, **b** Loadings for the third PLS dimension, and **c** corresponding scores. LSC, ASC, and PSC correspond to lateral, anterior, and posterior semicircular canals (in blue), Co refers to the cochlea (in red), and VW and CW to the vestibular and cochlear windows (in green). The orientation of the bony labyrinths is the same as in Fig. 2b. **d**–**f** Loadings and scores for the fourth PLS dimension. Source data are provided in the Source Data file.

(e.g., hearing range, sensitivity threshold at low and high frequencies). Nonetheless, as proxies our contextual variables sufficiently captured the adaptive signal. However, these four adaptive dimensions also carried a phylogenetic signal, indicating an overlap of adaptive and phylogenetic patterns (adaptive divergence) – they cannot be completely disentangled.

In contrast to our findings, various studies on mammals, birds, and reptiles have cast doubt on the role that ecology plays in shaping inner ear morphology[26,67–70]. Labyrinth shape sometimes shows a stronger relationship with skull size or shape than ecology or locomotion in birds, reptiles, and mammals[34,67–69,71,72]. Inner ear shape seems to track phylogeny well in various mammal groups, often interpreted as a result of neutral evolution[70,71,73,74]. However, the relative importance of adaptive versus neutral evolutionary processes in

explaining phenotypic variation depends on the taxonomic level and degree of environmental/behavioural diversity investigated. For instance, adaptive divergence into distinct ecological niches associated with differentiated phenotypes tends to be more visible at the level of major clades, such as *across* mammals rather than *within* subclades of mammals (e.g., ruminants, or groups at even lower taxonomic levels). In theory, our finding of morphological similarity among functionally and ecologically analogous taxa could also result from evolutionary stasis rather than convergence, which ultimately requires fossil evidence to distinguish. But given the deep phylogenetic divergence between afrotherians and their analogues as well as the consistent association of the shape patterns with the functional variables across clades, convergent evolution is a much more likely and parsimonious interpretation of our findings than evolutionary stasis.

While convergent evolution in traits such as body size, body proportions, and life history traits is well documented and easy to understand, it is less obvious how the vertebrate ear realises such a stunning evolvability. Nowhere else in the vertebrate skeleton are different functional units so close together, comprising the smallest bones of the skeleton that are jointly encapsulated in the surrounding temporal bone. Moreover, growth of the inner ear is already completed by the time of birth e.g., [75–78], which may further limit the ability to evolve. Le Maître et al.[17] speculated that this evolvability is a consequence of the genetic and developmental complexity of the mammalian ear. The evolutionary integration of several bones of the vertebrate jaw into the mammalian middle ear[1,79, and refs therein] not only improved and decoupled mastication and hearing[79,80], it also considerably increased the genetic, regulatory, and developmental complexity of the mammalian ear[13]. For instance, both the first and second pharyngeal arches give rise to the middle ear bones in mammals, whereas in other tetrapods only the second pharyngeal arch is involved[10,81]. This increase in the number of genetic and developmental factors, in turn, likely has increased the genetic variation of ear morphology that natural selection can act on[16,82,83]. Because of overlapping pleiotropic effects, it may also have increased the ability of different functional ear components to evolve independently[84–87]. In other words, it has increased the evolutionary degrees of freedom for an independent adaptation of the different functional units of the ear – the number of genetic and developmental knobs for natural selection to turn.

This increased evolvability of the ear and its sensory functions in mammals, relative to other vertebrates, may have contributed – among other factors – to the adaptive diversification of mammals into an astonishing range of niches, reflected by the high disparity to diversity ratios of mammals in general and of Afrotheria in particular. Both mammals and its afrotherian subclade have hearing abilities that far exceed those of other tetrapod clades, being capable of perceiving infrasonic to ultrasonic sounds[88,89]. Mammals also run the gamut of positional behaviours, which is mirrored almost entirely by Afrotherians alone (with the exception of flying and gliding). Our findings about convergent adaptations of inner ear shape in afrotherian and non-afrotherian mammals are in line with this expectation of high evolvability in the mammalian ear. A more explicit test of evolvability would require a comparison of between- and within-species variation of ear shape across mammals, birds, and reptiles, which is work in progress by our team.

## Methods
### Comparative sample
The sample consists of 20 Afrotheria and 20 non-afrotherian mammals considered to be their analogues. Analogues were identified based on their overall body morphology (e.g., the greater hedgehog tenrec, *Setifer setosus*, and the true hedgehog, *Erinaceus europaeus*), specific morphological traits such as relative hindlimb length (as in the case of the larger-bodied elephant shrews, e.g., *Rhynchocyon cirnei*, the mouse deer, *Tragulus javanicus*, and rabbit, *Oryctolagus cuniculus*), or locomotion and habitat (e.g., the subterranean golden moles and true moles, or sea cows and cetaceans). The analogues mostly comprise non-afrotherian placentals. We also included a marsupial and a monotreme as phylogenetic outgroups, each of which also serve as an analogue. Most afrotherians have more than one analogue and some species share the same analogue (e.g., the European mole is an analogue for the two species of golden mole as well as for the rice tenrec). All taxa used in the study are presented in Fig. 1 and Table 1. Elephants do not have any appropriate living analogues, but because they are very distinct extant afrotherians, we nonetheless included them in our study.

The sample was carefully composed based on the phylogenetic, morphological, and ecological disparity within Afrotheria, their appropriate analogues, and the ability of our team to acquire high-resolution micro-CT scans of the desired taxa (see below and our online repository for more details). The sample covers 22-25% of extant afrotherian diversity (based on a taxonomic count of 79-90 recognised species[90,91]) and all of living afrotherian disparity in terms of body size, ecology, lifestyles, and locomotor behaviour. Unfortunately, palaeontological specimens could not be included because ecology, positional behaviour, and overall morphology cannot be independently verified for fossil taxa. Furthermore, we used relatively strict criteria when selecting analogues by including only those taxa that were sufficiently analogous to the afrotherians in our sample in order to not dilute the signal of convergence.

### Contextual data
Relevant data on positional behaviour (locomotion and posture) and certain aspects of ecology were collected from the literature for each species. Data that directly measure auditory capacity could unfortunately not be included because common parameters of hearing ability (e.g., minimum/maximum perceived frequency, or frequency of greatest hearing sensitivity) were unknown for many species in the sample. However, habitat type (e.g., aquatic, subterranean, or ground-dwelling) carries information about the medium through which species detect sound and may thus indirectly also reflect basic aspects of hearing. Multiple aspects of ecology and positional behaviour were captured by 11 ordinal variables (Table 2, data provided in the online repository). Locomotion and postures of mammals vary extensively and they have been studied to varying degrees of detail for the species in our sample. For the purpose of this study, positional behaviour was measured on an ordinal scale. Habitat was thus captured by several of these ordinal variables rather than by a single categorical variable. Species were assigned a broad habitat type only as a visual aid for the PLS and PCA scores, but these categories were not used in any computational analyses. Species' mean body mass correlates with both ecology and locomotor behaviour across mammals, and so it was included as an additional contextual variable, measured on an interval scale.

Body mass was collected from PanTHERIA[92], and the other data were collected from mammalogical reference works[93–99], supplemented by other written sources[28,100] or, where necessary, videographic documentation of species' positional behaviour. We define agility as a combination of speed and manoeuvrability (adapted from Spoor et al.[28]).

### Phylogeny
The phylogenetic position of Afrotheria within Placentalia is not fully resolved, but molecular phylogenetic data supports Afrotheria as a monophyletic lineage separate from the more speciose clade Boreoeutheria (comprising Laurasiatheria and Euarchontoglires) e.g.,[42–44,101] (see also refs in Ref. 45). Most likely, Afrotheria is either sister to Boreoeutheria and Xenarthra combined ('Exafroplacentalia'), or it forms a clade together with Xenarthra ('Atlantogenata') which is sister to Boreoeutheria (reviewed in Zachos[102]). We followed the Atlantogenata hypothesis herein and employed the phylogeny in Fig. 1, downloaded from www.VertLife.org[103].

### Data acquisition
We landmarked a single bony labyrinth per species. As we are dealing with a sample of disparate taxa, most of which are separated by long evolutionary time spans, interspecific variation likely far exceeds within-species variation. Contextual data were also only available on a species level. Bony labyrinth morphology is only slightly sexually dimorphic[104,105], likely rendering dimorphism negligible compared to the disparity across species[106]. Similarly, asymmetry between left and right ears is very likely minute relative to the variation between species[107,108]. Hence, we did not consider within-species variation,

**Table 3 | Definition of the 13 anatomical landmarks and the 111 semilandmarks on the bony labyrinth**

| # | Type | Definition |
|---|------|-----------|
| | | **Anatomical landmarks** |
| 1 | LM | ASC ampulla to vestibule (midline) |
| 2 | LM | ASC ampulla to slender part of ASC (midline) |
| 3 | LM | CC to ASC & PSC (midline) |
| 4 | LM | CC to vestibule (midline) |
| 5 | LM | Slender part of PSC to ampulla (midline) |
| 6 | LM | PSC ampulla to vestibule (midline) |
| 7 | LM | Slender part of LSC to vestibule (midline) |
| 8 | LM | Slender part of LSC to ampulla (midline) |
| 9 | LM | LSC ampulla to vestibule (midline) |
| 10 | LM | Helicotrema |
| 11 | LM | Centre of the round window |
| 12 | LM | Posterior extremity of the axis of maximum elongation of the oval window |
| 13 | LM | Anterior extremity of the axis of maximum elongation of the oval window |
| | | **Semilandmarks** |
| 1-20 | SL (20) | Midline curve of the ASC: LM 1 → 2 → 3 (2 SL: ampulla, 18 SL: slender part) |
| 21-40 | SL (20) | Midline curve of the PSC: LM 6 → 5 → 3 (2 SL: ampulla, 18 SL: slender part) |
| 41-45 | SL (5) | Midline curve of the CCR: LM 3 → 4 |
| 46-65 | SL (20) | Midline curve of the LSC: LM 9 → 8 → 7 (2 SL: ampulla, 18 SL: slender part) |
| 66-105 | SL (40) | Midline curve of the cochlea: apical side of the RW → LM 10 |
| 106-111 | SL (6) | Outline of the oval window: LM 13 → 12 (inferior), then 12 → 13 (superior) |

Landmarks 1-10 and all semilandmarks are defined as in Gunz et al.[111], except for the different numbers of semilandmarks along each curve. For the semilandmarks, we indicate the order in which they are placed between anatomical landmarks. See Fig. 2 for visualisation.
*LM* anatomical landmark, *SL* semilandmark.

sexual dimorphism, and bilateral asymmetry; instead we aimed to maximise taxonomic, phylogenetic, and ecological breadth of the sample.

We downloaded microCT scans of 13 crania from MorphoSource (www.MorphoSource.org, Duke University), which had resolutions ranging from 7.88 to 82.67 μm (isometric voxel size), except for the aardvark (93*93*200 μm) and the opossum (59.6*59.6*132.0). These scans were originally contributed by the California Academy of Sciences, the Museum of Vertebrate Zoology at Berkeley, the Museum of Comparative Zoology, Harvard University, the Peabody Museum, Yale University, Ted Macrini, Robert Asher and colleagues[109], and Deborah Bird and colleagues[110]. We also generated new data in this work by scanning crania of 24 species curated at the Natural History Museum of Vienna, Austria, with resolutions ranging from 6.41 to 106.67 μm, as well as crania of three specimens from the Palaeontology Institute of the University of Vienna, Austria, with resolutions ranging from 9.62 to 24.95 μm. For each specimen, we virtually extracted the surface of one bony labyrinth using the software Amira (Thermo Fisher Scientific) version 2020.2.

To quantify the morphology of the bony labyrinth (Fig. 2, Table 3), we placed 13 anatomical landmarks (LM) and 111 semilandmarks (SL) on every surface model derived from the microCT scans. Apart from (semi)landmarks placed on the surface to quantify the centre of the cochlear window (LM 11) and the outline of the vestibular window (LM 12-13 and six semilandmarks), we positioned all landmarks on the centrelines of the semicircular canals, the common crus and the cochlea, computed using the AutoSkeleton module in Amira. Following

Gunz et al.[111], our anatomical landmarks included nine landmarks (LM 1-9) on the three semicircular canals: the proximal and distal extremities of each ampulla (LM 1, 2, 5, 6, 8, 9), the junction of the anterior and posterior semicircular canals at the common crus (LM 3), and the insertion of the common crus (LM 4) and the lateral canal (LM 7) on the vestibule. A final anatomical landmark (LM 10) was placed at the apex of the cochlea. The morphology of each canal was described by a curve comprising 20 semilandmarks, including two semilandmarks on the ampulla. We also placed five semilandmarks on the common crus, and 40 semilandmarks on the cochlea. In many species, the posterior segment of the lateral semicircular canal is partially fused with the ampulla of the posterior canal before entering the vestibule, thus forming a secondary common crus. In this case, we landmarked the lateral semicircular canal up to the vestibule as if it were separate from the ampulla, because the corresponding membranous ducts are not fused internally[76].

## Data analysis

In order to standardise for overall location, scale, and orientation of the landmark configurations, we performed a Generalised Procrustes Analysis (GPA) with mirroring allowed, because the sample comprised both left and right ears[112,113]. We minimised bending energy when sliding the semilandmarks[114,115]. Due to the huge variation in cochlear morphology, the magnitude of sliding per iteration was reduced by a factor of 0.1 and the number of iterations was increased accordingly. This variation also challenged the visualisation of statistical results as shape deformations because a change in the number of cochlear coilings is not achievable by a linear transformation; intermediate shapes may thus appear biologically unrealistic. Nonetheless, ordinations and PLS models appeared meaningful, despite suboptimal visualisations. We are not aware of a method to circumvent this problem; the present shape variation approaches the limits of what can be analysed by geometric morphometrics[116].

In the case of adaptive convergence, a given afrotherian species is expected to share more similarities in labyrinth morphology with analogues than non-analogues. In order to test this pattern of convergence, we computed all pairwise Procrustes distances between taxa and compared the average distances between four different types of pairs: (1) all pairs of afrotherians and their non-afrotherian analogues, (2) all pairs of afrotherians and non-afrotherian non-analogues, (3) all pairs of afrotherians with other afrotherians, and (4) all pairs of non-afrotherians with other non-afrotherians. The average Procrustes distance between pairs of afrotherians (type 3) and between pairs of non-afrotherians (type 4) were included so as to evaluate the signal of convergent evolution in relation to the signal of phylogenetic relatedness. Convergent evolution in bony labyrinth shape between Afrotheria and other eco-morphologically similar mammals can be inferred if the average pairwise distances of (1) is smaller than the average distances of (2), (3) and (4). Note that the elephant has no living analogue and was therefore only included in the computation of (2) and (3), and habitat categories (a classification for visual purposes in the PCA and 2B-PLS) were *not* used in the computation of Procrustes distances. See Table 1 for the detailed description of the pairs of analogues as well as Supplementary Table 1 and Supplementary Note 1 for further explanation of the different pairwise comparisons. Significance tests of the null hypothesis of equal average distances of the above four types were based on permutation tests that permuted the affiliation of species across Afrotheria-analogue pairs. We also calculated the Wheatsheaf index[55], which is the ratio of the average Procrustes distance between pairs of analogues to the average Procrustes distance between all species while statistically correcting for phylogenetic relatedness. Our approach above and the Wheatsheaf index are similar to other methods comparing morphological and/or phylogenetic distances for the study of convergent evolution[117–119].

To explore the morphological diversity of the bony labyrinth across species, we performed a principal component analysis (PCA) of the Procrustes shape coordinates (results are shown in Supplementary Figs. 1 and 2 and in Supplementary Note 2). We also performed a two-block partial least squares (2B-PLS) analysis to investigate the association between bony labyrinth shape and the 12 contextual variables[120–122]. The contextual variables were mean-centred and scaled to unit variance. We also conducted a phylogenetic 2B-PLS, based on phylogenetic generalised least squares (PGLS)[123] and the phylogeny in Fig. 1, in order to assess whether the associations between bony labyrinth shape and the contextual variables were mediated by the species' phylogenetic relationships (Supplementary Figs. 3–4 and Supplementary Note 3). Additionally, we performed leave-one-out and leave-two-out cross-validations to assess the robustness of the results to changes in sample composition (Supplementary Figs. 5–7 and Supplementary Notes 4 and 5).

All analyses were performed in R version 4.3.1[124] as well as in Wolfram Mathematica 12, with nearly identical results. Results presented here were obtained in R, and R scripts are provided in the online repository. We used the R packages ape 5.7-1[125], geomorph 4.0.6[126,127], Morpho 2.11[128] and phytools 2.0-3[129]. We generated scatter plots using the packages ggplot2 3.4.4[130] and ggrepel 0.9.4[131]. Visualisations of the shape changes and contextual loadings presented here were generated in Mathematica.

Finally, to evaluate to what extent the associations between bony labyrinth shape, ecology and positional behaviour are phylogenetically patterned, we quantified the phylogenetic signal of the PLS dimensions by means of Blomberg's $K$[132]. Phylogenetic signal, or the tendency for closely related species to resemble each other in ecology and/or morphology, is expected to be high when close relatives are more similar to each other than they are to distant relatives. Conversely, phylogenetic signal is low when trait values vary randomly across the phylogenetic tree or when distant relatives tend to resemble each other more than closely related species, as in evolutionary convergence e.g.,[132,133]. To compute Blomberg's $K$ for the shape and context scores along the PLS dimensions, we used the function 'phylosig' from the package phytools[129] and the phylogenetic tree, with branch lengths scaled to time, in Fig. 1.

This combination of methods for assessing the pattern of convergence and relating it to adaptive factors that may have shaped this pattern is, to our knowledge, a novel approach and one that lends itself well to geometric morphometric data, but it can also be applied to other continuous variables at an interval scale. It makes no assumptions about underlying evolutionary processes and evolutionary independence of traits (but see refs. 116,134 for the implicit assumptions underlying Euclidean and Procrustes distances).

### Reporting summary
Further information on research design is available in the Nature Portfolio Reporting Summary linked to this article.

## Data availability
All 3D surface models, the raw and slid landmark coordinates (not Procrustes-aligned), as well as the detailed sample composition, the table of contextual variables and the ultrametric phylogenetic tree generated in this study are available and freely accessible in the OSF repository [https://osf.io/9mtwh/]. Source data for Figs. 3 and 4, and all Supplementary Figures are provided in the Source Data file. Supplementary Figures, Tables, and Notes were uploaded directly as a single document (Supplementary Information) associated with the main text to the journal website. Source data are provided with this paper.

## Code availability
All R code and raw data files required to reproduce the analyses are available as supplementary data files in a GitHub repository [https://github.com/dalemaitre/Afrotheria], also available on Zenodo[135] and in the paper's OSF repository [https://osf.io/9mtwh/].

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

## Acknowledgements

This research was funded in whole or in part by the Austrian Science Fund (FWF) [grant https://doi.org/10.55776/P33736, to P.M., N.D.S.G., C.P., and A.L.M.]. For open access purposes, the authors have applied a CC BY public copyright license to any author-accepted manuscript version arising from this submission. This work was further supported by FWF grant number ESP 485 (N.D.S.G.) and by the Konrad Lorenz Institute for Evolution and Cognition Research (A.L.M.). The California Academy of Sciences, Museum of Vertebrate Zoology at Berkeley, as well as Museum of Comparative Zoology, Harvard University, and Peabody Museum, Yale University provided access to CT scans, the collection of which was funded by oVert TCN. Robert Asher provided access to data originally appearing in Asher et al.[109]. Deborah Bird provided access to data originally appearing in Bird et al.[110], the collection of which was funded by NSF Graduate Research Fellowship Program grant DGE-1144087. Ted Macrini provided access to other data, with data collection funded by NSF DEB-0309369 and data upload to MorphoSource funded by DBI-1902242. All the aforementioned data were downloaded from www.MorphoSource.org, Duke University.

## Author contributions

N.D.S.G., P.M. and A.L.M. designed the research, with input from F.E.Z. and C.P. A.L.M., C.P., F.H., G.B.M., V.W., and F.E.Z. collected scan and surface model data. N.D.S.G., F.H. and F.E.Z. collected contextual data, and N.D.S.G. acquired the phylogeny. F.H. and G.B.M. landmarked the data. N.D.S.G., P.M. and A.L.M. performed analyses and made figures. N.D.S.G., F.E.Z., P.M. and A.L.M. interpreted the results. N.D.S.G., P.M. and A.L.M. wrote the paper, and F.E.Z. contributed to the text.

## Competing interests

The authors declare no competing interests.
