## [Peer Review File · Nature Communications]

Convergent evolution in Afrotheria and non-afrotherians demonstrates high evolvability of the mammalian inner earReviewers' Comments:

Reviewer #1:

Remarks to the Author:

This is a manuscript that provides a 3D Geometric morphometric approach of the variations in mammalian inner ear morphology. The conclusion specially emphasizes convergent evolution between afrotherians and other placental mammals.

The manuscript is overall well written and of high scientific quality. The methods are sound and the results are well analysed and discussed. The discussion section provides an interesting overview of adaptations of inner ear traits and a discussion about the evolvability of the inner ear, which are quite speculative but of high interest to specialists, palaeontologists in particular.

The main weakness lies in the novelty.

Firstly, the methods (3D geometric morphometric of inner ear) is not particularly novel. See for example:

- Lebrun, R., De León, M.P., Tafforeau, P. and Zollikofer, C. (2010), Deep evolutionary roots of strepsirrhine primate labyrinthine morphology. *Journal of Anatomy*, 216: 368-380.

- and applied to afrotherians: Benoit J., Lehmann T., Vatter M., Lebrun R., Merigeaud S., Costeur L., and Tabuce R. (2015) Comparative anatomy and three dimensional geometric-morphometric study of the bony labyrinth of Bibymalagasia (Mammalia, Afrotheria). *Journal of Vertebrate Paleontology*, 35(3):e930043.

- Secondly, the main conclusion "that functional adaptation and convergence dominate over phylogenetic history" in placental mammals is not an unexpected result. Since the advent of DNA based phylogenetics, it has become quite clear that adaptation-drive convergent evolution was the main driver of mammalian phylogeny. Convergent evolution in the ear region of afrotherians has been discussed previously in Benoit J., Merigeaud S. and Tabuce R. (2013) Homoplasy in the ear region of Tethytheria and the systematic position of Embrithopoda (Mammalia, Afrotheria). *Geobios*, 46(5): 357-370.

- Thirdly, the other conclusion that "the mammalian ear has high evolvability" has already been well-established by Ekdale (2013, 2016, cited in the text).

Missing some crucial papers about convergent evolution in the Afrotherian ear: Benoit 2013 (embrithopods and convergence with elephants) / Benoit 2013 on sirenians (and convergence with cetaceans)

Overall, this is a high quality study with interesting results that are unquestionably worth publishing, but the rather expected results and overall lack of novelty, I think, make this work unsuitable for a highly competitive journal.

Reviewer #2:

Remarks to the Author:

Dear editors, dear authors,

I had the pleasure to review the article entitled “Convergent evolution in Afrotheria and non-afrotherians demonstrates high evolvability of the mammalian inner ear”. The article depicts the presence of an ecological signal and a phylogenetical signal across the mammal tree based on 40 species. Such study has never been done at this phylogenetic level. The dataset is medium comparing the last articles published in Nature Communications with around 200 species (Evers et al. 2022, Mennecart et al. 2022) or 130 in Scientific reports (Benoit et al. 2020). The first and most important ecological signal is the aquatic life with a reduction of the semi-circular canals. The second aspect is the fossorial behavior involving the angle between the anterior and the posterior canals and a narrow cochlea. The last PLS3 is about agility with relative expansion of the canals. The manuscript is well written and easy to follow.

Using the provided landmarks, I have tested your dataset using hierarchical analyses to see morphological proximities (.nex file resulting provided here). I was very surprised that *Amblysomus/Chrysochloris* cluster with *Talpa/Cryptomys* and also *Hemicentetes/Setifer/Tenrec* with *Erinaceus* that clearly indicating for sure something maybe ecological. I was unfortunately unable to observe other grouping for ecology but “just” phylogenetic natural clades, especially Afrotherians (hyrax, Chrysochloridae, Tenrecidae, Macroscélides, Sirenia). Then I am wondering if the sampling should be more extensive. Indeed, when including 20 species from the super-order Afrotheria, you see a clear phylogenetical signal. What would happen if you add more from the 5 other super-order considered in this study? This has also been demonstrated in Cetaceans that increasing the number of species, noise that was first interpreted as an ecological signal is lost. In the case of Cetaceans, only extreme divers (more than 2000 meters) have a convergent evolution of the cochlea (Parks et al. 2019).

One very interesting result in my opinion is about the cochlea. Actually, when looking at the cochlea, we can discriminate Afrotheria from other mammals based on the lateral view of the turns. But you are not considering the volume of the bony labyrinth here, even the cochlea itself actually if I follow correctly. You centered the line in the middle of the cochlea that goes approximately in the spiral canal (see Costeur et al. 2017). So, the shape of this neuronal portion may give ecological information that is great!

Unfortunately, I could find neither the R script nor the table including the numerical results. Then I was not able to test the results. I miss some analyses to be certain that they are testing the ecological signal and not something else. A Var.part analysis may help to solve the impact of each parameters on the results. Is there a difference if you are using the median instead of the mean (which is biologically more relevant)? What about the comparison with the phylogenetic tree? Can you please compare the phylogenetic distances in the way you are doing the morphological distances to be certain that your results are not link to a bias in the topology of your tree? Are there

enough specimens per categories considering the degree of freedom? I did not find reclassification scores to test the robustness in reclassification prediction.

The ecological aspect of the bony labyrinth has been several time challenged lately using large datasets (Benoit et al. 2020, Araujo et al. 2022, David et al. 2022, Evers et al. 2022, Mennecart et al. 2022) and soft tissues may be necessary to obtain ecological parameters (David et al. 2016). Metabolism (mammaliaforms), neutral evolution (ruminant), skull configuration (bird), and allometry (turtle) may be the most important factors that shape evolution after these authors. How would you consider your results in the light of these researches? Considering your 3 first PLS (the 4th has a p.value of more than 0.05 and not significant, right?), I would like to discuss and understand the results.

Among your outliers, your first one is the dolphin. I would advise to delete this specimen because your dataset is not normalized due to this specimen (or include transition forms from land to sea in cetacean lineage to normalize). Including the dolphin that is so different may attract and bias the ecologically/shape for the aquatics. Then after deleting this specimen, please retest your dataset to see if the reduction of the canals is still observed and not biased by this specimen.

Considering PLS2, you indicate the angle between the anterior and the posterior canals and a narrow cochlea for fossorial behavior. This is very interesting. The platypus is put as a semi-aquatic but also dig nests. Moreover, Mennecart et al. 2022 (and other papers of them), use these characters for phylogeny and family splitting in ruminants. Tragulidae are in graph 3F the closest to the vole. However, they have these morphological characteristics since at least the Early Miocene with different ecologies and environments depending of the species. They are not known to dig. Are all the ecological categories well-defined?

Considering PLS3, the agility with relative expansion of the canals has been challenged by Evers et al. (2022). They show that turtles have relatively larger canals than mammals, despite being less agile.

Why did you prefer to have in the text the analysis with phylogenetical component and in the supplementary the one that take into account for phylogeny?

Chrysochlorys and Ochotona are also outliers. Indeed their shape are strange. Are the entire bony labyrinths in open access to verify the original shape?

I will go now more specifically to some points of the MS

L35. Size, Costeur et al. 2017 demonstrated that the allometry of the BL is clearly minor. A shrew has a 5mm BL while a whale has a 3cm BL. This is related to the fact that this is the first organ to ossify around mid-gestation (Mennecart & Costeur 2016, Costeur et al. 2018, and others).

L37. The head posture has been challenged by Benoit et al. 2020. They found that the head posture is correlated to the phylogeny and the canals also, but the orientation of the canals cannot predict the head posture.

L53 “high disparity to diversity ratio” do you mean today? In comparison to what? To each super-order considered in the analysis? All mammal species?

L74 “assume high evolvability” who assume that? Considering the above mentioned papers, they

demonstrated a relatively low evolvability in comparison to other organs.

Figure 1 is difficult to read and we miss all the ecological parameters. Maybe you could add the 8 contextuels to this figure using colors instead of 1, 2, 3 right after the name in the phylogeny and under use the animal outlines in pair with their analogues forming the 12 groups of Afrotheres and analogues.

Figure 3 It seems they may be some problems in coding or specimens when looking at the shape of the cochlea (in red) in the various figures. Maybe this is again linked to the dolphin. In these figures, the extreme shapes are not displayed (-0.1 & +0.1). Please show the extreme values (-0.2 & +0.2).

L151-155. The p.value is not significant. Maybe you should delete this part.

L158 how different is it? What the impact of the phylogeny here?

L191-197 references are missing.

L198 instead of very small -> significantly smaller

L202 mammals have

L202 limited number of turns, how many?

L227-235 references are missing.

L240 the ossification is long before birth excepted for open structures that are considered in this study. There are studies on humans, pigs, dolphins, ruminants.

L243-246 references are missing.

L263-264 this has actually been done in Evers et al. 2022

L282 what is high resolution? Usually this is between 40 and 60 micro.

Thank you very much for your study and I am looking forward to see it published.

Best regards,

Bastien Mennecart

Here is the literature I am referring in the review:

Araújo, R., David, R., Benoit, J. et al. Inner ear biomechanics reveals a Late Triassic origin for mammalian endothermy. *Nature* 607, 726–731 (2022). <https://doi.org/10.1038/s41586-022-04963-z>

Benoit J., Legendre L., Farke A., Neenan J., Mennecart B., Costeur L., & Manger P.R. (2020). The lateral semicircular canal and head posture in “ungulate” mammals: implications on diet, behavior and paleobiological reconstructions. *Scientific Reports* 10, 19602. DOI 0.1038/s41598-020-76757-0

Costeur L. et al. (2017). Innervation of the cow’s inner ear derived from micro-computed tomography. *Proceedings Spie* doi: 10.1117/12.2276158

Costeur, L., Mennecart, B., Müller, B., & Schulz, G. (2019). Observations on the scaling relationship between bony labyrinth, skull size and body mass in ruminants. *Proceedings Spie*, 36. <https://doi.org/10.1117/12.2530702>

Costeur L., Mennecart B., Müller B., & Schulz G. (2019). Observations on the scaling relationship between bony labyrinth, skull size and body mass in ruminants. *Proceedings of SPIE* 11113, 1111313-1–1111313-10. DOI 10.1117/12.2530702.

David R., Bronzati M., & Benson R.J. Comment on “The early origin of a birdlike inner ear and the evolution of dinosaurian movement and vocalization”. *Science* 376, 6600 (2022).

<https://doi.org/10.1126/science.abl6710>

David, R. et al. Assessing morphology and function of the semicircular duct system: introducing new in-situ visualization and software toolbox. *Sci. Rep.* 6, 32772; doi: 10.1038/srep32772 (2016).
Evers, S.W., Joyce, W.G., Choiniere, J.N. et al. Independent origin of large labyrinth size in turtles. *Nat Commun* 13, 5807 (2022). <https://doi.org/10.1038/s41467-022-33091-5>
Mennecart B., Dziomber L., Aiglstorfer M., Bibi F., DeMiguel D., Fujita M., Kubo M.O., Laurens F., Meng J., Métais G., Müller B., Rios M., Rössner G.E., Sanchez I.M., Schulz G., Wang S., & Costeur L. (accepted). Ruminant inner ear shape records 35 million years of neutral evolution. *Nature Communications*. <https://www.nature.com/articles/s41467-022-34656-0> (06.12.2022)
Mennecart, B., & Costeur, L. (2016). Shape variation and ontogeny of the ruminant bony labyrinth, an example in Tragulidae. *Journal of Anatomy*, 229(3), 422–435. <https://doi.org/10.1111/joa.12487>
Park T., Mennecart B., Costeur L., Grohé C., & Cooper N. (2019). Convergent evolution in toothed whale cochleae. *BMC Evolutionary Biology* 19, 195. DOI 10.1186/s12862-019-1525-x

Reviewer #3:

Remarks to the Author:

The manuscript by Grunstra and colleagues reports on inner ear shape (the bony labyrinth) in Afrotherians and non-afrotherians across a broad range of species occupying different ecological niches. The use a multivariate approach to evaluate shape of the bony labyrinth as it relates to 12 contextual variables that capture multiple aspects of ecology and positional behaviour. The authors conclude that afrotherians resemble their analogues more closely in labyrinth shape than they do each other or non-afrotherian non-analogues, and that the inner ear exhibits high evolvability in mammals. This is an interesting analysis, however, some concerns regarding the data analysis and interpretation remain.

Major points:

1. The first result presented is the Procrustes distances between all pairs of afrotherians and non-afrotherians. These data are discussed as averages: "The average Procrustes distance between afrotherians and their non-afrotherian analogues was 0.238, whereas the average distance between afrotherians and (non-afrotherian) non-analogues was 0.281." From these data, the authors conclude that "functional adaptation dominates over phylogenetic history." I'm assuming that Procrustes distance was measured for each category (ground-dwelling, aquatic, etc) and then the authors made an average. However, these data should be presented per category in table form. It seems clear from Figs. 3,4 that some categories are more disparate than others and even may not be statistically different. In that regard, it would be notable which functional categories overlap and which are distinct.
2. The data in Figs. 3,4 and Fig. S3 do not appear to show analyses for significance or variance. In the text describing these figures, shapes are described morphologically, but not quantitatively. The authors do discuss statistics for phylogenetic signals, but such statistical analyses are missing from the adaptive signals. Some of the adaptive dimensions are clearly better at separating the

groups, and this could be discussed if it is statistically supported. The ground-dwelling groups occupy a lot of shape space, which could also be a point of discussion.

3. Only one image of the bony labyrinth is shown. It would be more convincing to see a representative image of a bony labyrinth from each of the major eco-types. This is especially important given the amount of discussion in the text relevant to the shapes.

4. The authors conclude that 1) the mammalian ear exhibits “high evolvability,” 2) that this is surprising because the bones are small and close together, and 3) the evolvability is a direct consequence of the increased genetic and developmental complexity of the mammalian ear compared to other vertebrates.” All of these conclusions seem a bit tenuous to me. “Evolvability” as a character trait is not evaluated. I think the authors are suggesting that the bony labyrinth exhibits a lot of morphological variation compared to other structures, but they have not actually quantified that. The authors also imply that because the mammalian middle ear consists of more elements than the avian middle ear and/or because it consists of both first and second pharyngeal arch components that it is more genetically complex than the avian middle ear, but in fact, there is no evidence for that. Avian mandibles consist of more bones than the mammalian mandible, but I don’t think anyone would suggest the avian mandible is genetically more complex than the mammalian mandible. It may rather be that the pharyngeal arches (notably PA1 and PA2) have developmental mechanisms that contribute to evolvability which may impact any or all of their derivatives which are quite variable among vertebrates. Since the authors do not present any genetic or developmental data, nor do they evaluate other characters within mammals or ear structures in any other clades, these comparative statements seem unwarranted.

Minor points:

1. In lines 31-34, the authors state: “We previously proposed that this increased heterogeneity and developmental complexity of the mammalian ear bestows it a higher evolvability, i.e., an enhanced capacity for adaptive evolution, and that this increase in evolvability likely contributed to the evolution of the spectacular disparity in mammalian body plans and ecological niches.” This also seems a bit of an overstatement. As mentioned above, developmental complexity of the mammalian ear is not shown, but linking evolvability of the middle ear would to disparity of body plans is a big leap.

REVIEWER COMMENTS

We are grateful to the reviewers, especially reviewer 2, for taking the time to provide us with critical feedback, which has substantially improved our manuscript. We have responded to all reviewers' points (see below) and revised our manuscript accordingly, much to its improvement.

In our responses to reviewer 2, we carefully explain why we have opted not to expand our sample and why it does not affect the validity of our results below. We have also incorporated part of this justification in the manuscript itself.

Furthermore, we have included all of our R code in our online repository (in Github linked to our cited OSF repository, see Data Availability). Although originally all analyses were carried out in the software Mathematica and replicated in R, we opt here to supply the R code rather than the Mathematica code, because the former will serve a much larger portion of the scientific community. In our initial submission, we reported the results obtained in Mathematica, but now that we supply the R code we have changed the numerical results in our manuscript to those obtained in R (if and when they differ from Mathematica's), to be consistent and transparent. If results obtained in the different softwares differed, they only did so very slightly and merely in a numerical sense, i.e. none of the patterns, interpretations and concomitant conclusions changed.

Revisions to the manuscript are highlighted in yellow. We also quote line numbers in our responses below.

Reviewer #1:

This is a manuscript that provides a 3D Geometric morphometric approach of the variations in mammalian inner ear morphology. The conclusion specially emphasizes convergent evolution between afrotherians and other placental mammals.

The manuscript is overall well written and of high scientific quality. The methods are sound and the results are well analysed and discussed. The discussion section provides an interesting overview of adaptations of inner ear traits and a discussion about the evolvability of the inner ear, which are quite speculative but of high interest to specialists, palaeontologists in particular.

The main weakness lies in the novelty.

Firstly, the methods (3D geometric morphometric of inner ear) is not particularly novel. See for example:

- Lebrun, R., De León, M.P., Tafforeau, P. and Zollikofer, C. (2010), Deep evolutionary roots of strepsirrhine primate labyrinthine morphology. *Journal of Anatomy*, 216: 368-380.
- and applied to afrotherians: Benoit J., Lehmann T., Vatter M., Lebrun R., Merigeaud S., Costeur L., and Tabuce R. (2015) Comparative anatomy and three dimensional geometric-morphometric study

of the bony labyrinth of Bibymalagasia (Mammalia, Afrotheria). Journal of Vertebrate Paleontology, 35(3):e930043.

Response: Clearly, you are right that the application of 3D geometric morphometrics to inner ear morphology is not novel in itself, nor did we claim so. The novelty of our study lies in the way we investigated and quantified evolutionary convergence in afrotherians and other mammals, namely based on a single multivariate summary statistic of convergence (ratio of average Procrustes distances) in the sample along with an exploratory approach (2B-PLS) to estimate the shape features that drive this convergence. To our knowledge this is novel. We extended the explanation of this in the main manuscript (lines 107-125, and 404-410) as we obviously did not explain this properly before.

- Secondly, the main conclusion "that functional adaptation and convergence dominate over phylogenetic history" in placental mammals is not an unexpected result. Since the advent of DNA based phylogenetics, it has become quite clear that adaptation-drive convergent evolution was the main driver of mammalian phylogeny. Convergent evolution in the ear region of afrotherians has been discussed previously in Benoit J., Merigeaud S. and Tabuce R. (2013) Homoplasy in the ear region of Tethytheria and the systematic position of Embrithopoda (Mammalia, Afrotheria). Geobios, 46(5): 357-370.

Response: We agree with the reviewer that signals of adaptation are to be expected in highly functional traits such as the inner ear, especially at higher taxonomic levels such as among major placental mammalian clades. We briefly review this literature in the second paragraph of the Introduction (L. 58-71). However, our study was not aimed at demonstrating convergent evolution in mammals *per se* or in specific afrotherian species or subclades, as this is well-established (we added further references to the fossil Afrotheria literature by Benoit, Tabuce and colleagues, lines 73-74, 88-94 and 97-101). Rather, we aimed to quantify the overall pattern and degree of convergence in the bony labyrinth between Afrotheria and other placentals as this had not been done before at this taxonomic/phylogenetic level (as also pointed out by reviewer 2). In the scientific literature, the inner ear is often argued to be useful for inferring both functional behaviors (e.g. auditory capacity, locomotion) and phylogenetic relationships. With regard to the latter, it is often implied that the inner ear is a more reliable indicator of ancestry than of adaptation and that this mainly reflects neutral evolution. Whereas sometimes functionally adaptive and phylogenetic signals overlap and the two cannot be disentangled, cases of convergent evolution are a notable exception. Our sample of afrotherians and distantly related placentals therefore lends itself well to comparing these two signals. We believe that the quantification of bony labyrinth shape (dis)similarity within the Afrotheria clade vs. between afrotherians and other placentals is novel in this context.

- Thirdly, the other conclusion that "the mammalian ear has high evolvability" has already been well-established by Ekdale (2013, 2016, cited in the text).

Response: Ekdale has indeed shown numerous adaptive convergences in the bony labyrinth morphology of mammals with similar ecologies (e.g., marine carnivora, aquatic mammals), agility or hearing physiology, which is in agreement with our findings, but Ekdale and others have not put this in the context of the modern evolvability literature. Moreover, whereas Ekdale measured ear morphology and *qualitatively* compared it across species in relation to their ecology, locomotor behaviour or hearing abilities, we *statistically quantified* the association between ear morphology and contextual variables. We believe that the evolvability context is relevant and interesting here

because the consistent inner ear adaptations in different mammalian lineages appears surprising given (1) the tight spatial and development constraints in the inner ear (e.g., as opposed to adaptations in overall body form, limb morphology, etc.), and (2) that eco-behavioral signals are weak or absent in other tetrapod groups, such as birds and “reptiles”. We hypothesised in Le Maître et al. (2020) that the higher developmental complexity through the transformation of jaw bones into middle ear ossicles crucially contributes to this evolvability in mammals. Therefore we decided to present the current study within this context.

Missing some crucial papers about convergent evolution in the Afrotherian ear: Benoit 2013 (embriothopods and convergence with elephants) / Benoit 2013 on sirenians (and convergence with cetaceans)

Response: We agree that these references are relevant to our work and have added them (L. 71, 73-74 and 97-101). Thank you for pointing them out.

Overall, this is a high quality study with interesting results that are unquestionably worth publishing, but the rather expected results and overall lack of novelty, I think, make this work unsuitable for a highly competitive journal.

Response: We tried to better explain the novelty of our methodology and results in the manuscript (please see also our responses above).

Reviewer #2:

Dear editors, dear authors,

I had the pleasure to review the article entitled “Convergent evolution in Afrotheria and non-afrotherians demonstrates high evolvability of the mammalian inner ear”. The article depicts the presence of an ecological signal and a phylogenetical signal across the mammal tree based on 40 species. Such study has never been done at this phylogenetic level. The dataset is medium comparing the last articles published in Nature Communications with around 200 species (Evers et al. 2022, Mennecart et al. 2022) or 130 in Scientific reports (Benoit et al. 2020). The first and most important ecological signal is the aquatic life with a reduction of the semi-circular canals. The second aspect is the fossorial behavior involving the angle between the anterior and the posterior canals and a narrow cochlea. The last PLS3 is about agility with relative expansion of the canals. The manuscript is well written and easy to follow.

Response: Thank you for the positive feedback on our manuscript. Indeed, our sample size appears small compared to some other studies. But the papers you quoted studied much more speciose clades and/or had different research questions (e.g. tracing the evolutionary history of the inner ear in turtles or ruminants through time). Afrotheria is a notably species-poor taxon with a maximum of 90 recognized living species (but as low as 79 [Wilson & Reeder 2005]). The disparity mostly exists between larger clades, all of which we sampled and usually by more than one species (e.g. several hyraxes, several tenrecs), covering 22-25% of extant afrotherian diversity depending on the taxonomy used. The analogues in our current sample were carefully selected based on the level and nature of their “analogy” to afrotherians. Adding additional analogues might weaken the convergence signal because they are insufficiently analogous to afrotherians or they are essentially duplicates of taxa we have already included (see further comments on sample composition in our response below.)

Using the provided landmarks, I have tested your dataset using hierarchical analyses to see morphological proximities (.nex file resulting provided here). I was very surprised that Amblysomus/Chrysochloris cluster with Talpa/Cryptomys and also Hemicentetes/Setifer/Tenrec with Erinaceus that clearly indicating for sure something maybe ecological. I was unfortunately unable to observe other grouping for ecology but “just” phylogenetic natural clades, especially Afrotherians (hyrax, Chrysochloridae, Tenrecidae, Macroscélides, Sirenia). Then I am wondering if the sampling should be more extensive. Indeed, when including 20 species from the super-order Afrotheria, you see a clear phylogenetical signal. What would happen if you add more from the 5 other super-order considered in this study? This has also been demonstrated in Cetaceans that increasing the number of species, noise that was first interpreted as an ecological signal is lost. In the case of Cetaceans, only extreme divers (more than 2000 meters) have a convergent evolution of the cochlea (Parks et al. 2019).

Response: Thank you for your interesting comments and for taking the time to look at our data. Because adaptive, phylogenetic and other signals may overlap, comparing ecologically analogous species one by one or in terms of simple clusters in shape space can be ineffective. This is the reason we introduced a new approach that tries to estimate these signals at the level of the full sample. Apparently, this was not well described (as argued also by other reviewers), and thus we added additional explanations (L. 107-125, L. 404-410).

As mentioned above, our sample size is comparatively small but we capture a large portion of extant afrotherian diversity (species numbers) and all of their disparity. Adding further non-afrotherians to our present sample will not necessarily lead to a clearer signal, because we already took care to sample species that were suitably analogous (usually in both an ecological and morphological sense). Species that are “ecomorphological duplicates” to our existing analogs are very close relatives of the species we already used (e.g. a hare and a rabbit, or two delphinids), which would increase the sample size but also the phylogenetic signal. Adding more species could even *dilute* the convergent signal if they are ecologically less similar than the analogous taxa we already used. Adding fossils to our sample would not have been helpful since their ecology, hearing capacities and positional behaviour are not sufficiently known and cannot really be independently verified.

We think our research question about convergent evolution in the mammalian inner ear using Afrotheria and “analogous” mammals as a test case can be adequately addressed with our current sample composition. Nonetheless, we thank the reviewer for raising this concern because it has given us the opportunity to demonstrate the robustness of our findings with respect to sample size and composition. Specifically, we performed a series of resampling studies of both the Procrustes distance comparisons and the PLS analysis. Results proved to be very stable when omitting one or two species (jackknifing). This was true even when omitting the dolphin, which the reviewer suggested is an outlier and could bias our results. We added a short paragraph on this at the end of the Results section (L. 197-207) and a detailed report in the Supplementary Information (Supplementary Fig. 5-7 and related text).

One very interesting result in my opinion is about the cochlea. Actually, when looking at the cochlea, we can discriminate Afrotheria from other mammals based on the lateral view of the turns. But you are not considering the volume of the bony labyrinth here, even the cochlea itself actually if I follow correctly. You centered the line in the middle of the cochlea that goes approximately in the spiral canal (see Costeur et al. 2017). So, the shape of this neuronal portion may give ecological information that is great! Unfortunately, I could find neither the R script nor the table including the numerical results. Then I was not able to test the results. I miss some analyses to be certain that they are testing the ecological signal and not something else. A Var.part analysis may help to solve the impact of each parameters on the results. Is there a difference if you are using the median instead of the mean (which is biologically more relevant)?

Response: We have uploaded the R scripts (on GitHub and on the OSF repository). Unfortunately, we do not understand what you mean with (presumably) an analysis of variance to assess the impact of each parameter. The relative contribution of each contextual variable is assessed as part of the PLS analysis. We have added extensive jackknife resampling studies to assess the robustness of our results, which indicate robust loadings of the contextual parameters (see the Supplementary Fig. 7 and related Supplementary Notes). Similar results are obtained when comparing median instead of mean Procrustes distances. We added this information to the manuscript (L. 140-141 and Supplementary Fig. 5-6 and related Supplementary Notes).

What about the comparison with the phylogenetic tree? Can you please compare the phylogenetic distances in the way you are doing the morphological distances to be certain that your results are not link to a bias in the topology of your tree?

Response: By “the comparison of the phylogenetic tree” and “a bias in our phylogenetic tree” we assume you are concerned that the ratio in our Procrustes distances between types of pairs may be biased if afrotherians are more closely related to their analogues than to their non-analogues. The latter is not the case, which can be seen in the phylogenetic tree in Fig. 1. Firstly, the ratio of Procrustes distances between types of pairs showed that afrotherians are morphologically more similar to mammals to which they are clearly phylogenetically more *distantly* related (smaller average Procrustes distance), compared to fellow afrotherians (larger average Procrustes distance). Secondly, when comparing morphological similarity between afrotherians to analogues vs. non-analogues, there is also no bias deriving from our tree topology, because afrotherians are equally distantly related to both the analogues and non-analogues from a given clade (e.g. Xenarthra, or Boreoeutheria) and this holds true for all clades.

Are there enough specimens per categories considering the degree of freedom? I did not find reclassification scores to test the robustness in reclassification prediction.

Response: We are not sure what you mean here. We did not use categories in any computations, we only assigned habitat categories as a visual aid in the PLS and PCA figures (now provided in the online OSF repository https://osf.io/9mtwh/?view_only=88a550a637f84615b5666d69278309d4, in the file "Grunstra_et_al_contextual.csv"). However, apparently this was not sufficiently clear before, and so we have added a sentence in the captions of Figs. 3 and 4, supplementary Fig. 1 and 3, and in L. 327-330 to make this clearer. Therefore, there is no computation related to classification, nor any prediction of category for a given taxon, so the number of cases per habitat category is irrelevant.

The ecological aspect of the bony labyrinth has been several time challenged lately using large datasets (Benoit et al. 2020, Araujo et al. 2022, David et al. 2022, Evers et al. 2022, Mennecart et al. 2022) and soft tissues may be necessary to obtain ecological parameters (David et al. 2016). Metabolism (mammaliaforms), neutral evolution (ruminant), skull configuration (bird), and allometry (turtle) may be the most important factors that shape evolution after these authors. How would you consider your results in the light of these researches? Considering your 3 first PLS (the 4th has a p.value of more than 0.05 and not significant, right?), I would like to discuss and understand the results.

Response: Clearly, also other factors than those studied by us have contributed to mammalian inner ear evolution. However, findings by other authors about the strength of an ecological signal in one group cannot automatically be extrapolated to other taxonomic groups. Studies that focus on clades like turtles (Evers et al. 2022), cetaceans (Costeur et al. 2018, Park et al. 2019), bats (Davies et al. 2013) and ruminants (Mennecart et al. 2022) look at groups with less morphological, ecological or locomotor *intra-group* disparity compared to Afrotheria. In this case, other factors (e.g. spatial constraints within the cranium) or even neutral evolution may increase in relative importance. That is not the case in our study of Afrotheria and other mammals, in which we found clear correlations between inner ear shape, ecology and locomotion, and we detected shape similarity between distant relatives. David et al. (2022) evaluated functional signals in non-mammalian tetrapod groups, but it is conceivable that tetrapod groups separated by deep splits, such as “reptiles” and mammals, which differ in a number of other important aspects (physiology, labyrinth morphology, etc.), show patterns of phenotypic variation that require different explanations. Similarly, Araújo et al. (2022) found a relationship between labyrinth shape and body temperature across synapsid clades, but this relationship was much less clear within (placental) mammals (their Fig. 2).

Indeed, when studies found phylogenetic signal to be strong in bony labyrinth morphology, this was often at lower taxonomic levels and/or within clades of closely related species that have a similar ecological niche or locomotor behaviour (e.g. within humans, among cetaceans, or among deer). We argue that whether morphology (predominantly) reflects neutral evolution is a non-generalizable question that depends on the trait, the taxonomic group, and the taxonomic level, as the relative importance of selection vs. drift obviously varies widely across the tree of life and in response to environmental conditions. And in any case, although strong neutral evolution leads to phylogenetic signal, phylogenetic signal does not necessarily and only reflects neutral evolution, as phylogenetic divergence may be adaptive.

Regarding soft tissue anatomy, it is well established that the cross-section area of the semicircular ducts, a measurement not available without soft tissues, is an important biomechanical parameter for the vestibular system (e.g. Muller 1999, David et al. 2016). Our findings demonstrate that, even if we are missing some functionally relevant information of the inner ear, the functional (and presumably adaptive) signal is strong enough in our sample to be detected.

Lastly, body mass (allometry) is one of our contextual variables but seems to play a role only for PLS dimension 4. Indeed, the fourth dimension has a $p > 0.05$, but we decided to show it nonetheless because it has a clear interpretation and the resampling led to stable replicates. After all, because of the relatively small sample and the non-random sample composition, the p -values should not be overinterpreted.

Here are references not cited in the main text we are referring to:

Araújo, R., David, R., Benoit, J., Lungmus, J. K., Stoessel, A., Barrett, P. M., ... & Angielczyk, K. D. (2022). Inner ear biomechanics reveals a Late Triassic origin for mammalian endothermy. *Nature*, 607(7920), 726-731.

Davies, K.T., Maryanto, I. & Rossiter, S.J. Evolutionary origins of ultrasonic hearing and laryngeal echolocation in bats inferred from morphological analyses of the inner ear. *Front Zool* 10, 2 (2013). <https://doi.org/10.1186/1742-9994-10-2>

David, R., Stoessel, A., Berthoz, A., Spoor, F., & Bennequin, D. (2016). Assessing morphology and function of the semicircular duct system: introducing new in-situ visualization and software toolbox. *Scientific reports*, 6(1), 32772.

Among your outliers, your first one is the dolphin. I would advise to delete this specimen because your dataset is not normalized due to this specimen (or include transition forms from land to sea in cetacean lineage to normalize). Including the dolphin that is so different may attract and bias the ecologically/shape for the aquatics. Then after deleting this specimen, please retest your dataset to see if the reduction of the canals is still observed and not biased by this specimen.

Response: The dolphin may be a statistical outlier in the sample, but we do observe the same pattern of reduced semicircular canals and a large cochlea in the two other strictly aquatic mammals (manatee and dugong). We think this reflects true biological variation. However, even when removing the dolphin, we still find the same pattern along PLS 1 that contrasts aquatic and non-aquatic species (see the resampling results in the SI: supplementary figure 7). Furthermore, although semi-aquatic taxa do not display such an extreme labyrinth morphology, they nonetheless tend towards the strictly aquatic pattern (see Fig. 3a-c), supporting PLS 1 as a reliable functional

pattern. As the dolphin is the only strictly aquatic non-afrotherian taxon we have in our sample - and there are no other strictly aquatic clades - we prefer to keep it in our main analysis.

Considering PLS2, you indicate the angle between the anterior and the posterior canals and a narrow cochlea for fossorial behavior. This is very interesting. The platypus is put as a semi-aquatic but also dig nests. Moreover, Mennecart et al. 2022 (and other papers of them), use these characters for phylogeny and family splitting in ruminants. Tragulidae are in graph 3F the closest to the vole. However, they have these morphological characteristics since at least the Early Miocene with different ecologies and environments depending of the species. They are not known to dig. Are all the ecological categories well-defined?

Response: As mentioned before, the habitat categories used in the PLS plots (e.g. Fig. 3c and 3f) are only used for visualization purposes; they are not used as a variable in any computations. Furthermore, many species show a mixed repertoire of environments or substrates that they use and hence these categories are not a comprehensive reflection of taxa's ecological strategies. To capture variation in the latter in our analyses, we used the 12 contextual variables (shown in e.g. Fig. 3a and 3d). Regarding the habitat categories as a visual tool, however, we opted to display the platypus as semi-aquatic because we felt this described a more salient aspect of its locomotion and ecology than the fact that it digs and sleeps in a burrow. Although we classified the vole species in our sample (*Microtus arvalis*) as subterranean, it is also ground-dwelling a substantial portion of the time (influencing its context scores along the Y axis in Fig. 3c), which might explain why it bears morphological similarities to other ground-dwelling taxa (see its shape scores along the X axis in Fig. 3c). Lastly, ecological similarity and phylogenetic distance/relatedness may show overlap in some clades (e.g. ruminants, or within Afrosoricida) but less so in others, and the degree to which this is the case will also be sample/clade-dependent.

Considering PLS3, the agility with relative expansion of the canals has been challenged by Evers et al. (2022). They show that turtles have relatively larger canals than mammals, despite being less agile.

Response: This is interesting indeed. We think it is well possible that adaptation of bony labyrinth shape has occurred differently in divergent clades such as turtles and mammals when these clades differ in many important aspects such as overall body morphology, positional behavior and, importantly, physiology (incl. endo- vs. ectothermy), which may affect the biomechanics of the vestibular system. These aspects include for example the ionic composition of the endolymphatic and the perilymphatic fluids, and hence the ability for electromotile hair cells to generate a signal, or the cross-section area and position of the membranous duct in the bony semicircular canal and therefore biomechanical properties of the vestibular system, as well as the mechanism controlling head stabilisation through the vestibulo-collic reflex which may be limited by the characteristic turtle shell.

A stronger semicircular canal ellipticity in turtles relative to mammals may also play a role, because deviation from circularity tends to decrease canal sensitivity, especially for extremely elliptic canals (McVean, 1999). Small semicircular canals are less sensitive than large canals, and this effect is much stronger for elliptic canals. Therefore, having very large semicircular canals could be a way to compensate for high ellipticity (or, alternatively, having very elliptic canals to compensate for their large size), as observed in squamates (Goyens, 2019). Relatively large but strongly elliptical canals are thus not as sensitive as relatively large and rounder canals.

Here are reference not cited in the main text we are referring to:

Goyens, J. (2019). High ellipticity reduces semi-circular canal sensitivity in squamates compared to mammals. *Scientific Reports*, 9(1), 16428.

Why did you prefer to have in the text the analysis with phylogenetical component and in the supplementary the one that take into account for phylogeny?

Response: We presented only one of the two types of analysis in the main text for the sake of space. We chose to present the analysis with the phylogenetic component in the main text because the convergent signal was more obvious: despite *not* taking phylogenetic relationships into account, whereby one may expect phylogenetic relatedness to “swamp” any signs of convergence, the pattern of convergent evolution already came through quite clearly (especially for aquatic and fossorial lifestyles). We carried out the phylogenetic PLS to verify if the same convergent ecomorphological associations remained after accounting for phylogeny, which was the case.

Chrysochloris and *Ochotona* are also outliers. Indeed their shape are strange. Are the entire bony labyrinths in open access to verify the original shape?

Response: We have now uploaded the surface models of all bony labyrinths used in our study to the online repository (https://osf.io/9mtwh/?view_only=88a550a637f84615b5666d69278309d4). We carefully checked the segmentation and there were no anomalies, so the surface models reliably reflect the microCT scans. Indeed, our *Chrysochloris* specimen has an unusual shape of the oval window although the rest of the labyrinth shape appears normal. Nonetheless, *Chrysochloris* is similar to the other golden mole *Amblysomus* in the PLS results, supporting the notion that its bony labyrinth shape is not an artefact. We do not see where/when *Ochotona* may be an outlier in either the PCA plots (SI) or the PLS plots (main manuscript).

I will go now more specifically to some points of the MS

L35. Size, Costeur et al. 2017 demonstrated that the allometry of the BL is clearly minor. A shrew has a 5mm BL while a whale has a 3cm BL. This is related to the fact that this is the first organ to ossify around mid-gestation (Mennecart & Costeur 2016, Costeur et al. 2018, and others).

Response: We are not sure what you mean by “minor allometry”. In the papers you are referring to, a clear negative allometric relationship is observed between bony labyrinth and skull size / body mass (e.g. Costeur et al. 2019). We guess you mean that absolute size variation (and not allometry) is minor because of early ossification? We agree that in an absolute sense, especially compared to skull or body dimensions, labyrinth size does not vary a lot. However, in a sample of Afrotheria, Benoit et al (2015) found radii ranging from 0.61 mm (*Chrysochloris asiatica*) to 4.09 mm (*Orycteropus afer*) for the posterior semicircular canal. This factor of ~7 between both values demonstrates that, even if the variation is not large in absolute value, size variation is not completely negligible for the bony labyrinth.

Regarding allometry, it was also shown by others that, at least in some clades, allometry is an important component of shape variation (e.g. del Rio et al. 2021 for Platyrrhine primates, and Le Maître et al. 2023 for Papionin primates, among others). Nonetheless, we have rephrased our sentence to talk about “relative size” rather than absolute size (L. 58).

L37. The head posture has been challenged by Benoit et al. 2020. They found that the head posture is correlated to the phylogeny and the canals also, but the orientation of the canals cannot predict the head posture.

Response: In this particular passage (now L. 58-61) of our manuscript, we do not make specific statements with regard to the orientation of the lateral semicircular canal (LSC) and head posture. However, we agree that the Benoit et al. (2020) study shows compelling evidence that, based on the loose relationship between head posture and lateral semicircular canal (LSC) orientation, at least in ungulate mammals, an “evolutionary adaptation” of bony labyrinth morphology to posture in mammals may be a bit of an overstatement. We have rephrased the sentence, and added references that at least suggest links between ear morphology and body posture in hominoid primates.

L53 “high disparity to diversity ratio” do you mean today? In comparison to what? To each super-order considered in the analysis? All mammal species?

Response: Here we talk of the four main placental mammal clades, so the groups of reference are Xenarthra, Archontoglires and Laurasiatheria (we have added this clarification in L. 73-74). Xenarthra have even lower diversity than Afrotheria in terms of species numbers, but even taking into account fossil taxa, disparity is also lower than in Afrotheria (for example, neither fossorial nor fully aquatic xenarthrans are known to have existed). Arguably the only group of the four main clades rivalling Afrotherian disparity is Laurasiatheria (which includes fossorial, aquatic and, with the bats, even flying mammals), but while there are 80 to 90 Afrotherian species, there are more than 2000 Laurasiatherians.

L74 “assume high evolvability” who assume that? Considering the above mentioned papers, they demonstrated a relatively low evolvability in comparison to other organs

Response: We were referring to the hypothesized high evolvability of the mammalian ear relative to the ear in other vertebrate clades, as discussed in the first paragraph of the introduction and in Le Maître et al. 2020, which we now expanded on (L. 41-42 and L. 49-52). We have also modified our original sentence and added the reference to our previous paper for clarification (L. 102).

Figure 1 is difficult to read and we miss all the ecological parameters. Maybe you could add the 8 contextualls to this figure using colors instead of 1, 2, 3 right after the name in the phylogeny and under use the animal outlines in pair with their analogues forming the 12 groups of Afrotheres and analogues.

Response: All 12 (not 8) contextual variables are already provided as supplementary data in the online data repository (https://osf.io/9mtwh/?view_only=88a550a637f84615b5666d69278309d4, link provided in the “Data Availability” section). As these 12 parameters yield a 12x40 data matrix, we do not see an easy way to present these data in Fig. 1. In case you are referring to the basis of analogy (e.g. *Potamogale velox* and *Lutra lutra* are similar in their semi-aquatic lifestyle and overall morphology), we have added an explanation using examples in the figure legend of Fig. 1, and it is explained further in the Methods text and Table 1.

Figure 3 It seems they may be some problems in coding or specimens when looking at the shape of the cochlea (in red) in the various figures. Maybe this is again linked to the dolphin. In these figures, the extreme shapes are not displayed (-0.1 & +0.1). Please show the extreme values (-0.2 & +0.2).

Response: The problem here is not a particular specimen but the variance in the number of coils in the cochlea between species. An increase or decrease in the number of coils, as represented by PLS 2, cannot properly be achieved by a linear transformation of the shape coordinates, as intermediate shapes do not correspond to realistic geometries. This is why the visualization looks somewhat odd. We carefully investigated this problem and are convinced that the ordinations and PLS models are nonetheless meaningful, even though the visualizations are suboptimal. Unfortunately, we are not aware of a method to circumvent the problem; here we seem to approach the limits of phenotypic variation that can be analysed with geometric morphometrics, We added a comment of this issue in the Methods section (L. 384-390).

L151-155. The p.value is not significant. Maybe you should delete this part.

Response: Although it is true that PLS 4 failed to reach statistical significance, we prefer to keep the brief description of this dimension in because we believe the component to be meaningful and interpretable (non-agile vs. agile species), even if it explains a small portion of the total covariance. In fact, the latter helps drive the non-significant p-value, but one should not put too much emphasis on *p*-values when interpreting data, especially when sample sizes are not that large.

L158 how different is it? What the impact of the phylogeny here?

Response: As can be seen from the comparison of the regular PLS (Figs. 3 and 4) and the phylogenetic PLS (supplementary Figs. 3 and 4), the association between labyrinth shape and the eco-locomotor variables along PLS 1 stays roughly the same, i.e. a contrast between aquatic vs. more ground-dwelling, pronograde species, and the labyrinth shape pattern also stays very similar. Consecutive PLS dimensions also remain similar, though the association with particular ecologies (e.g. fossoriality) are sometimes differently distributed across the PLS dimensions. For instance, in the phylogenetic PLS fossoriality is associated with a similar labyrinth shape as before, but along phylo PLS 3 instead of phylo PLS 2. We have added these details to our description of the phylogenetic PLS in the Supplementary Information).

It is not surprising that the results differ per se; weighting the data by the (inverse of the) phylogenetic covariance matrix reduces the variance - and thus the covariance between blocks. The reduction of the total squared covariance between ecology and ear shape can also partly be explained by the fact that phylogeny mediated the ecomorphological associations before (in the regular PLS). This is not surprising given that several closely related taxa in our sample have similar ecologies and similar bony labyrinth shapes. As part of the phylogenetic adjustment, such species are artificially made less similar in both morphology and ecology, and therefore the total covariance between blocks is reduced. However, as argued also in the next section (“phylogenetic signals in labyrinth shape”), it cannot be ruled out that closely related taxa with similar ecologies have shared inner ear adaptations. In other words, the impact of phylogeny cannot a priori be assumed to be due to neutral evolution; adaptive and phylogenetic signals may overlap.

L191-197 references are missing.

Response: We are not sure where references might be missing. The paragraph in L. 191-197 you refer to (now L. 209-215) basically summarizes our main findings.

L198 instead of very small -> significantly smaller

Response: Done, although we chose to replace “very small” with “particularly small [...] in relation to other species”, because “significantly” might wrongly convey the idea that we tested for statistical significance. (L. 216)

L202 mammals have

Response: Here we mean to refer to our obtained results, and hence we used the past tense.

L202 limited number of turns, how many?

Response: Between 1.5 and 2, except for the walrus (~2.5 turns) and the otter (~3 turns). We have explicated this in the manuscript (L.221).

L227-235 references are missing.

Response: In this paragraph (now L.242-251), we again recap our findings and so there are no references to insert.

L240 the ossification is long before birth excepted for open structures that are considered in this study. There are studies on humans, pigs, dolphins, ruminants.

Response: We agree that our phrasing was a bit ambiguous, so in the sentence we replaced “completed around birth” by “completed by the time of birth”, which acknowledges the fact that growth can be finished long before birth (L. 266). Our point was only to highlight the fact that *after* birth, there is no more growth of the inner ear. You are right in pointing that open structures can still grow after ossification, but our landmark scheme was based nearly exclusively on closed structures, except for the windows. We did not study structures that continue to grow after birth, such as the cochlear aqueduct or the endolymphatic sac.

L243-246 references are missing.

Response: This sentence stems from the reference cited in the previous sentence (Le Maître et al. 2020), and is further developed in the following sentences. We added more precise references regarding the integration of vertebrate jaw bones into the mammalian ear and the decoupling of mastication and hearing functions (now L. 268-271).

L263-264 this has actually been done in Evers et al. 2022

Response: Evers et al. (2022) indeed briefly compared turtles, mammals, birds and “reptiles”, but not in the sense we intended here. As developed in the article in which we proposed the “evolvability hypothesis” (Le Maître et al. 2020), one macroevolutionary way to test this hypothesis would be to quantify the association between ear morphology and ecological variables for (placental) mammals and e.g. modern birds (clades of roughly similar age). If the ear is more evolvable in mammals than in other vertebrate groups, then this association should be stronger in mammals, compared to other vertebrate clades of similar age. In their paper, Evers et al. (2022) investigated the association between labyrinth morphology and diverse environmental and behavioural factors in turtles, and then they compared labyrinth size relative to skull dimensions in different amniote clades, but they did not compare the strength of the association between ear morphology and ecology for the different clades.

L282 what is high resolution? Usually this is between 40 and 60 micro

Response: It depends on the species (smaller species require better resolution), but we aimed at having voxel sizes below 80 microns to avoid "voxelized" volumes. See the section "data acquisition" (L. 354-360) and supplementary data for more details. The highest resolution is 6.41 microns (house mouse), the lowest 106.67 microns (walrus) .

Thank you very much for your study and I am looking forward to see it published.

Best regards,

Bastien Mennecart

Response: Thank you again for your helpful feedback and engagement!

Reviewer #3:

The manuscript by Grunstra and colleagues reports on inner ear shape (the bony labyrinth) in Afrotherians and non-afrotherians across a broad range of species occupying different ecological niches. They use a multivariate approach to evaluate shape of the bony labyrinth as it relates to 12 contextual variables that capture multiple aspects of ecology and positional behaviour. The authors conclude that afrotherians resemble their analogues more closely in labyrinth shape than they do each other or non-afrotherian non-analogues, and that the inner ear exhibits high evolvability in mammals. This is an interesting analysis, however, some concerns regarding the data analysis and interpretation remain.

Response: Thank you for your constructive feedback and comments on our manuscript. Below, we address the concerns you raised. In particular, we clarified the questions regarding "eco-types", which were not defined for any quantitative analysis, but merely as a visual aid for the PCA and PLS plots. We also explain better what we mean by evolvability and the increase in genetic and developmental complexity of the mammalian ear below and in the manuscript (for more details, see our point-by-point responses).

Major points:

1. The first result presented is the Procrustes distances between all pairs of afrotherians and non-afrotherians. These data are discussed as averages: "The average Procrustes distance between afrotherians and their non-afrotherian analogues was 0.238, whereas the average distance between afrotherians and (non-afrotherian) non-analogues was 0.281." From these data, the authors conclude that "functional adaptation dominates over phylogenetic history." I'm assuming that Procrustes distance was measured for each category (ground-dwelling, aquatic, etc) and then the authors made an average. However, these data should be presented per category in table form. It seems clear from Figs. 3,4 that some categories are more disparate than others and even may not be statistically different. In that regard, it would be notable which functional categories overlap and which are distinct.

Response: We did not compute Procrustes distances by habitat category. Habitat groups were only used as a visual aid in the presentation of the PLS scores (and PCA scores in the SI). We have added explanations in the main text (L. 327-330) and in figure captions (Figs. 3-4, and Supplementary Figs. 1, 3-4) to avoid confusion between this “pseudo”-variable and the 12 contextual variables used in the PLS and phylogenetic PLS analyses. This was not sufficiently clear before.

Procrustes distances were computed for all pairwise comparisons between species. We then computed the average Procrustes distance across (1) all pairs of afrotherians and their analogues, (2) all pairs of afrotherians and non-analogues, (3) all pairs of afrotherians with other afrotherians, and (4) all pairs of non-afrotherians with other non-afrotherians. Habitat categories were thus not used in the calculation or interpretation of Procrustes distances or indeed for any computations. The rationale behind our Procrustes distance approach is that *on average*, ecologically analogous species should resemble each other more than non-analogous species in the case of convergent evolution (see Table 1 and Fig. 1 for which taxa we consider analogous to each afrotherian species). We have extended the description of this rationale in the introduction (L. 119-121) and in the methods section (L. 393-395 and Supplementary Table 1 and accompanying Supplementary Notes). Note that because now we provide the R scripts (which are more likely to be re-used by the scientific community than Mathematica scripts), we replaced the values originally provided in the main text, which had been obtained with the software Mathematica. However, the results obtained in R and Mathematica were nearly identical and supported the same interpretation of the data and associated conclusions.

2. The data in Figs. 3,4 and Fig. S3 do not appear to show analyses for significance or variance. In the text describing these figures, shapes are described morphologically, but not quantitatively. The authors do discuss statistics for phylogenetic signals, but such statistical analyses are missing from the adaptive signals. Some of the adaptive dimensions are clearly better at separating the groups, and this could be discussed if it is statistically supported. The ground-dwelling groups occupy a lot of shape space, which could also be a point of discussion.

Response: Thank you for the feedback and giving us the opportunity to clarify. We did in fact present statistical support for the four adaptive dimensions obtained in the PLS, namely by providing significance levels (L. 148). However, we have now also added the proportion of the summed squared covariances explained by each of the PLS dimensions (L. 149-150, 153-154, 161 and 165-166).

As explained above and in the figure captions, the five habitat groups did not serve as the basis for any statistical comparison. They were designed only as a *visual* aid, without a strict definition for each group (the variable “habitat groups” is now provided in the OSF online repository, in the file "Grunstra_et_al_contextual.csv"). In this context, the group "ground-dwelling" is just a melting pot of all species that tend to dwell on the ground and do not fit in any other groups (strictly aquatic, semi-aquatic, (semi)arboreal, subterranean). Therefore, given the taxonomic and morphological heterogeneity of this "group", it is not surprising that it occupied more space in morphospace than the other ones, especially because it comprised more species. Because these habitat groups discretize species' ecologies that are in reality more complex and multi-dimensional, we did not use these groups for any computation.

3. Only one image of the bony labyrinth is shown. It would be more convincing to see a representative image of a bony labyrinth from each of the major eco-types. This is especially important given the amount of discussion in the text relevant to the shapes.

Response: As explained above, the "eco-types" are only broadly defined (especially the "ground-dwelling"), and are used only as a visual aid in the scatter plots. The description of the PLS results and discussion are done in reference to the loadings of the contextual variables, not to these "eco-types", (although the negative shape scores are solely represented by the strictly aquatic eco-type due to the remarkable morphology of aquatic taxa along PLS1). Because the "eco-types" were not used in any of the computations, and only rarely in the description of the results, we think it is not relevant to show shape deformations specific to individual habitat groups, especially to avoid confusion regarding the data we used for our interpretations. Anyway, the reader can get a good idea of the corresponding patterns by looking at the shape patterns with highest covariance in the PLS analysis, which roughly correspond to some of these "eco-types". For inspection of individual inner ear shapes, we also added the 3D surface models of all specimens on the online repository.

4. The authors conclude that **(1) the mammalian ear exhibits “high evolvability,” (2) that this is surprising because the bones are small and close together, and (3) the evolvability is a direct consequence of the increased genetic and developmental complexity of the mammalian ear compared to other vertebrates.**” All of these conclusions seem a bit tenuous to me. “Evolvability” as a character trait is not evaluated. I think the authors are suggesting that the bony labyrinth exhibits a lot of morphological variation compared to other structures, but they have not actually quantified that. The authors also imply that because the mammalian middle ear consists of more elements than the avian middle ear and/or because it consists of both first and second pharyngeal arch components that it is more genetically complex than the avian middle ear, but in fact, there is no evidence for that. Avian mandibles consist of more bones than the mammalian mandible, but I don’t think anyone would suggest the avian mandible is genetically more complex than the mammalian mandible. It may rather be that the pharyngeal arches (notably PA1 and PA2) have developmental mechanisms that contribute to evolvability which may impact any or all of their derivatives which are quite variable among vertebrates. Since the authors do not present any genetic or developmental data, nor do they evaluate other characters within mammals or ear structures in any other clades, these comparative statements seem unwarranted.

Response: We appreciate your feedback and respond to each point below.

- (1) **“The mammalian ear exhibits high evolvability”**: By “evolvability”, we mean the ability to evolve (in an adaptive way). Here we postulate (as we did previously in Le Maître et al. 2020) that the ear is more evolvable in mammals relative to the ear of other vertebrates (which we now clarify in L. 102, 280-281). Although we do not directly test this hypothesis by comparing mammals to another vertebrate clade in the present paper, we do draw on theoretical and empirical work to support it as a reasonable hypothesis (and we have added more details in the Introduction, L.41-42, 49-52), and our results of adaptation and convergent evolution are at least consistent with this hypothesis. There is no one, standard measure of evolvability *per se*, but we have shown several adaptive convergences in the mammalian inner ear, which is an indirect, macroevolutionary way of evaluating evolvability of the mammalian ear, which may be expected to be highly constrained given its short developmental window and the spatial constraints imposed on it given its location in the cranium. Indeed, if the ear did not have the ability to evolve in an adaptive way, we would not find any associations between ear morphology and contextual variables - and it would be even less likely to find similar morpho-functional associations for two separate groups of mammals.

- (2) **“... this is surprising because the bones are small and close together”**: here we recapitulated part of our argument originally discussed in our previous paper in which we proposed the “evolvability hypothesis” (Le Maître et al. 2020). The ear bones are packed together in the limited space of the temporal bone, which imposes a spatial constraint on them, thus limiting their ability to vary morphologically (“spatial packing” of the inner ear). This is probably why many studies find a substantial scaling effect of ear components to the surrounding cranium. On top of this, smaller bones tend to have, just by a scaling effect, a lower size variance than larger bones, likewise leading to a reduced ability to evolve different sizes.
- (3) **“the evolvability is a direct consequence of the increased genetic and developmental complexity of the mammalian ear compared to other vertebrates”**: we did not conclude this based on the present study, but there is good theoretical and empirical support for this based on the definition of “complexity” in the evolvability literature and what is known about the development and involved gene expression patterns of the middle ear of mammals versus other tetrapod clades. We reviewed this evidence in our 2020 paper in which we proposed the “evolvability hypothesis” (Le Maître et al. 2020) First of all, the mammalian middle ear includes two extra ossicles compared to other tetrapods, as well as a tympanic ring. Furthermore, two pharyngeal arches (PA1 and PA2) are involved in mammalian ear development, not just a single arch (PA2) as in non-mammalian amniotes. The involvement of an additional pharyngeal arch means that cell lineages from additional origins contribute to ear development (e.g. neural crest cells from the mandibular migration), and that more diverse gene expression patterns are involved (including downstream) since these differ between each PA (Takechi & Kuratani 2010, Fuchs & Tucker 2015). (We do not mean to imply that the developmental genetics *within* each arch is more complex in mammals.) The ear as a functional unit is therefore genetically and developmentally more complex in mammals compared to other tetrapods (or even all vertebrates), as mammal ear development involves the first pharyngeal arch *in addition to* the other embryonic structures (i.e. PA 2) also “used” in other clades. This increases the number of “knobs” that evolution can turn, i.e. ways and directions in which heritable and adaptive variation can be generated, and it thus enables new ways for the mammalian ear to evolve. In other words, it possesses higher evolvability. We have added additional text to our Introduction to make this clearer (L. 41-42 and L. 49-52), and we added several references to the genetic and developmental work supporting our argumentation (also cited in Le Maître et al. 2020) (e.g. in L. 49-52 and L. 269-271).

Minor points:

1. In lines 31-34, the authors state: “We previously proposed that this increased heterogeneity and developmental complexity of the mammalian ear bestows it a higher evolvability, i.e., an enhanced capacity for adaptive evolution, and that this increase in evolvability likely contributed to the evolution of the spectacular disparity in mammalian body plans and ecological niches.” This also seems a bit of an overstatement. As mentioned above, developmental complexity of the mammalian ear is not shown, but linking evolvability of the middle ear would to disparity of body plans is a big leap.

Response: Of course we are a bit speculative here, and certainly adaptation of the ear is not the sole factor that contributed to the adaptive diversification of mammals. We have rephrased this part to be clearer and less strong (L. 54-57, see also L. 281-283.). Nonetheless, we think it is not unlikely that the colonisation of new niches by mammals would have facilitated - and be itself facilitated by

- morphological adaptations in the vestibular and auditory systems of the ear, extending the hearing ranges of mammals and contributed to the evolution of different positional behaviors. The capacity to generate such adaptive variation is evolvability, and enhanced evolvability of the ear would have increased the potential to develop/evolve new hearing ranges. As part of the expansion into novel ecological niches, morphological changes elsewhere in the body certainly also occurred, furthering successful colonisation of new niches and potentially paving the way toward other new niches, thus contributing to the evolution of disparate body plans over time.

As explained above, evidence of developmental complexity of the mammalian ear, relative to other amniotes, was extensively reviewed in the cited paper (Le Maître et al. 2020 and refs. therein), as well as all arguments supporting our claim that this evolvability of the ear might have enabled the evolvability of mammals as a clade.

Reviewers' Comments:

Reviewer #1:

Remarks to the Author:

In my opinion and given the response of the authors to my previous comments, the manuscript is acceptable as it stands.

Note that line 94: inline reference to (Benoit et al. 2015; Buckley 2013) should be superscripted.

Reviewer #2:

Remarks to the Author:

Dear authors, dear editors,

The authors did not take into considerations all the major comments of the 3 reviewers. Then I agree with reviewer 1 that the manuscript should be rejected.

Best regards

Reviewer #3:

Remarks to the Author:

The revised manuscript by Grunstra and colleagues is substantially altered to improve clarity and reproducibility- including R scripts for future analyses. Detailed author responses satisfy my previous concerns. This current version will be a nice addition to the literature.

Reviewer #4:

Remarks to the Author:

Overall impressions

This study is an exploratory treatment of morphological convergence between afrotherian mammals and putative analogous mammals outside Afrotheria. The results begin to address one or two of the predictions of the hypothesis of high evolvability in the mammal ear (Le Maître et al., 2020). In general, the present manuscript is written clearly. The statistical techniques are appropriate to the questions and are approached carefully. The relevance of the findings to mammal evolution, to inner ear function, and to the debate about evolvability is discussed thoroughly. I primarily have questions for the authors (including on aspects of the statistical techniques), suggestions for additional citations, and thoughts about ear development. The noteworthiness of this study relates to both methods and results. The authors use statistical measures that are otherwise commonplace pieces of many comparative works based on geometric morphometrics (distance and two-block partial least squares) but link them together with a set of contextual, ecological variables to explore the presence and strength of homoplastic patterns. The authors conclude that, indeed, there is morphological convergence in the inner ear of afrotherian

and other mammals, and that the axes on which shape variation primarily falls are closely tied to ecology. The total workflow is unique, though I think the manuscript should cite earlier examples of studies that used morphological distances to assess convergence.

Overall, this manuscript is thought-provoking and well written. More pointed thoughts follow, as do line-by-line comments and references. Please note that some of my thoughts from my line-by-line comments are summarized in a generalized fashion in the sections that immediately follow this introduction.

Sample size

As other reviewers noted and the authors discussed in their rebuttal, the study's sample size is modest, relative to recent studies in Nature Communications that asked broadly similar questions, used broadly similar methods, and/or worked in the same organ system (e.g., Arbour et al., 2019; Singh et al., 2021; Mennecart et al., 2022). The authors defend the size of their sample, suggesting that any signals of convergence might be weakened if additional species were included. I looked for other studies that made similar arguments, but most references I could find relate to the effects of morphological noise on reconstructing phylogenies (in which larger taxonomic samples are often desired) (e.g., Gaubert et al., 2005; Revell et al., 2008), rather than exploring convergence. Wouldn't it be possible to expand the sample and still detect convergence if using a tanglegram (e.g., Zelditch et al., 2017; Stange, et al., 2018) and C-measures (Stayton, 2015)?

That all said, although the authors do not lay out a detailed history of why the taxa in their sample are analogs, the authors state upfront that there are a priori reasons to view afrotherians as being convergent with other mammals. I note that the authors did not set out with the goal to explicitly test whether convergence in afrotherians and other mammals happens more frequently than expected by chance. Answering that question presumably would require a larger sample and different methods (mentioned above and discussed below). But given the a priori presumption of convergence, this sample, which by all appearances adequately addresses the breadth of morphological and ecological disparity in afrotherians, seems to be of adequate size to answer the questions posed by the authors in the present study.

Methods

The authors take a multivariate exploratory approach to quantify convergence between afrotherians and other mammals. The primary components of this approach are, first, a single summary statistic (a ratio of average Procrustes distances) calculated from a geometric morphometric analysis of shape, and second, an exploration of shape space (a two-block partial least squares regression analysis) with respect to contextual, ecological variables. The authors test the robustness of their findings with a jackknife analysis (i.e., removing different combinations of species to assess stability of the results). The logical flow of the analytical techniques is straightforward and the results relatively easy to interpret.

That said, the process of using average Procrustes distances to assess convergence resembles the Wheatsheaf index (Arbuckle et al., 2014), which tests for convergence using a ratio of Euclidean distances that accounts for phylogeny. Wheatsheaf was based on comparisons of Euclidean distances that were used in a morphometric context like the present study (Winemiller, 1991; Muschick et al., 2012). I am surprised that one or more such studies are not referred to here. Granted, the distances employed in the present study are Procrustes, so that is certainly an

innovation relative to those other comparisons (which use Euclidean distances). However, the authors should cite one or more studies that used ratios of distances to assess morphometric convergence, as they are forerunners of the approach the authors use. Since the Procrustes distance is a number that could presumably be plugged into the Wheatsheaf formula, would the authors consider calculating a formal “Wheatsheaf” index (i.e., incorporating phylogeny) based on Procrustes distances for this sample?

I also was curious why C-measures were not deployed in this study to assess convergence, since I presume the ratio of Procrustes distances, like the Wheatsheaf index, might capture stasis and distinctiveness (Stayton, 2015). C-measures can quantify strength of convergence, which could be compared across pairs of analogues. The C-measures also can be a component of tests for whether convergence happens more often than expected by chance (as in Zelditch et al., 2017). Were C-measures not used because there was not an explicit test whether convergence in inner-ear form is more frequent than is expected by chance in mammals?

Figures and Tables

Generally the figures and tables are appropriate and support the text. I encourage the authors to update the color schemes of their figures for accessibility (i.e., for readers with vision disabilities). For figures 3 and 4, the red and green colors look identical under color-deficient proof settings. More specifically, the 3-D reconstructions of vestibular and cochlear shapes look identical in color, and the arboreal and subterranean datapoints look identical in color. I will leave any comments on individual figures and tables in a separate section below.

Line-by-line comments

Lines 47-48: I’m not sure what this statement adds. Is the implication that the pinna and ear canal vary by species more than the soft tissues of external ears of non-mammalian tetrapods do? If so, that point is not clear. Or is the point that only mammals evolved a pinna and ear canal? That may be so, but other groups of tetrapods have evolved complex specializations of the outer ear (or non-ear specializations that aid hearing). Birds have no pinna made of skin and connective tissue, but owls and many diurnal raptors have the facial disc (Knudsen & Konishi, 1979; von Campenhausen & Wagner, 2006), which varies among species, is deformable by musculature, and aids sound collection. Among owls, the skin of the outer ear (as well as the bone, in some cases) also varies by species and can aid sound collection due to asymmetry. Birds also have specialized feathers covering the ear that don’t impede sound conduction (Rival, 2005) and presumably serve a protective function. In crocodylians, the soft tissues surrounding the outer ear develop into a muscular set of scaly flaps that close to protect the deeper structures during submersion (“ear flaps”, Shute & Bellairs, 1955).

Lines 50-57: How would the increase in overall developmental complexity of the middle ear impact the vestibular system? This topic is glossed over here, and the citations do not discuss mechanisms that would translate that increase in complexity to the vestibular system. It makes sense that increased developmental complexity of the middle ear (from an extra pharyngeal arch) could release constraints on the middle ear and allow for more morphological variation in that

system. Given that the middle ear ossicles are directly linked with the cochlea, it also is plausible that the auditory portion of the inner ear could be impacted by the additional evolutionary “knobs and dials”. Still, the inner ear derives from a separate embryonic structure. Why would we expect middle-ear changes to impact the semicircular canals, which are not auditory organs?

Line 75: “comprising only few species”

I’d encourage the authors to write the approximate count.

Lines 150-170: Several sentences in this section are grammatically a little hard to follow because of overuse of commas. I know there are space limitations, but I recommend breaking these results into smaller sentences for ease of reading.

Line 163: I suggest removing “subterraneous” (or at least changing to “subterranean”, which is used everywhere else in the manuscript). I think “fossorial” alone gets the point across, given that Table 2 states that subterranean habits are captured by the “fossoriality” locomotor mode.

Line 165: What does “curvature” mean in this context? The narrowness or tightness of the coil, or something else? It’s not clear.

Line 183: “too” is unnecessary because there is an “also” earlier in the sentence.

Line 191: “even” is unnecessary to understand the sentence.

Lines 209-210: I suggest writing “Convergent adaptations” (plural rather than singular) or explicitly using “homoplasy” or “pattern” at the beginning of the sentence. Some readers might think it’s circular to say that convergent adaptation (singular) is the most convincing evidence of adaptive evolution because “adaptation” could refer to the process or the resulting homoplastic pattern.

Lines 213-214: Consider removing “were able to” and just writing “We identified”. It’s simpler and saves a few words.

Lines 216-220: “particularly” doesn’t seem applicable in the framing of the updated sentence. The sentence reads like the size of aquatic animals’ (as a group) canals is relative to others (as a group), so they would just be small, not particularly so. All else being equal, relatively smaller canals will be less sensitive, so the subsequent discussion about being aquatic would still be supported without the emphasis. Also, the sentence's readability would benefit from being broken into two or three sentences.

Lines 404-410: The method (including permutation tests) deployed here sounds very similar to the Wheatsheaf index, minus phylogeny. The only difference I can see is that the distance used is Procrustes rather than Euclidean. If there is a difference that I have missed, could the authors please explain? I’ve read through Arbuckle et al. (2014), and the explanation of the Wheatsheaf workflow sounds just like the present study, just with a different kind of distance. Also, the C-measures (Stayton, 2015) are amenable to geometric morphometric data, as in Zelditch et al.

(2017) and Stange et al. (2018). C-measures do not make assumptions about evolutionary process.

Figures

Figure 1: It looks like the whiskers from the silhouette of *Sorex araneus* are duplicated and overlap the taxon name (only at its appearance near the bottom of the image).

Figure 2: It is a little challenging to see the landmarks in part A. The landmarks overlap with each other quite a lot because of the three-dimensionality of the structure. I don't know if there is an easy fix for that problem.

Figures 3 and 4: These are very nice images. Apart from my suggestion to update the color palette, these figures communicate the results well.

Tables

Table 1: Out of curiosity, why are rodents like the kangaroo mouse not “microcursorial”? Are they too specialized for leaping/bounding?

Table 2: Why include a “fully orthograde” state if none of the sampled taxa are fully orthograde?

Literature Cited

Arbour, J.H., Curtis, A.A. & Santana, S.E. Signatures of echolocation and dietary ecology in the adaptive evolution of skull shape in bats. *Nat Commun* 10, 2036 (2019).
<https://doi.org/10.1038/s41467-019-09951-y>

Arbuckle, K., Bennett, C.M. and Speed, M.P. (2014), A simple measure of the strength of convergent evolution. *Methods Ecol Evol*, 5: 685-693. <https://doi.org/10.1111/2041-210X.12195>

Philippe Gaubert, W Chris Wozencraft, Pedro Cordeiro-Estrela, Géraldine Veron, Mosaics of Convergences and Noise in Morphological Phylogenies: What's in a Viverrid-Like Carnivoran?, *Systematic Biology*, Volume 54, Issue 6, December 2005, Pages 865–894,
<https://doi.org/10.1080/10635150500232769>

Knudsen, E.I., Konishi, M. Mechanisms of sound localization in the barn owl (*Tyto alba*). *J. Comp. Physiol.* 133, 13–21 (1979). <https://doi.org/10.1007/BF00663106>

Le Maître A, Grunstra NDS, Pfaff C, Mitteroecker P. Evolution of the Mammalian Ear: An Evolvability Hypothesis. *Evol Biol.* 2020;47(3):187-192. doi: 10.1007/s11692-020-09502-0. Epub 2020 May 27. PMID: 32801400; PMCID: PMC7399675.

Mennecart, B., Dziomber, L., Aiglstorfer, M. et al. Ruminant inner ear shape records 35 million years of neutral evolution. *Nat Commun* 13, 7222 (2022). <https://doi.org/10.1038/s41467-022-34656-0>

Muschick, Moritz et al. (2012). Convergent Evolution within an Adaptive Radiation of Cichlid Fishes. *Current Biology*, Volume 22, Issue 24, 2362 - 2368

Liam J. Revell, Luke J. Harmon, David C. Collar, Phylogenetic Signal, Evolutionary Process, and Rate, *Systematic Biology*, Volume 57, Issue 4, August 2008, Pages 591–601, <https://doi.org/10.1080/10635150802302427>

Rival, F. (2005). Auricular diseases in birds. In 8th European AAV Conference (p. 333-339).

Shute, C.C.D. and Bellairs, A.d. (1955), The external ear in Crocodilia. *Proceedings of the Zoological Society of London*, 124: 741-749. <https://doi.org/10.1111/j.1469-7998.1955.tb07813.x>

Singh, S.A., Elsler, A., Stubbs, T.L. et al. Niche partitioning shaped herbivore macroevolution through the early Mesozoic. *Nat Commun* 12, 2796 (2021). <https://doi.org/10.1038/s41467-021-23169-x>

Stange, M., Aguirre-Fernández, G., Salzburger, W. et al. Study of morphological variation of northern Neotropical Ariidae reveals conservatism despite macrohabitat transitions. *BMC Evol Biol* 18, 38 (2018). <https://doi.org/10.1186/s12862-018-1152-y>

Stayton CT. The definition, recognition, and interpretation of convergent evolution, and two new measures for quantifying and assessing the significance of convergence. *Evolution*. 2015 Aug;69(8):2140-53. doi: 10.1111/evo.12729. Epub 2015 Aug 6. PMID: 26177938.

von Campenhausen, M., Wagner, H. Influence of the facial ruff on the sound-receiving characteristics of the barn owl's ears. *J Comp Physiol A* 192, 1073–1082 (2006). <https://doi.org/10.1007/s00359-006-0139-0>

Winemiller, KO. Ecomorphological Diversification in Lowland Freshwater Fish Assemblages from Five Biotic Regions. *Ecological Monographs*, Vol. 61, No. 4 (Dec., 1991), pp. 343-365

Zelditch, M.L., Ye, J., Mitchell, J.S. and Swiderski, D.L. (2017), Rare ecomorphological convergence on a complex adaptive landscape: Body size and diet mediate evolution of jaw shape in squirrels (Sciuridae). *Evolution*, 71: 633-649. <https://doi.org/10.1111/evo.13168>

REVIEWER COMMENTS

Reviewer #1:

In my opinion and given the response of the authors to my previous comments, the manuscript is acceptable as it stands.

Note that line 94: inline reference to (Benoit et al. 2015; Buckley 2013) should be superscripted.

Response: Thank you for your positive evaluation and we are glad that we have satisfied your previous concerns. We have corrected the inline reference (now L. 100).

Reviewer #2 (Remarks to the Author):

Dear authors, dear editors,

The authors did not take into considerations all the major comments of the 3 reviewers. Then I agree with reviewer 1 that the manuscript should be rejected.

Best regards

Response: We are sorry to hear that you feel not all of your comments were taken into consideration. Summarizing our previous point-by-point responses to your first, overall-positive review, we justified our sample size and composition on the basis that we cover ~25% of extant afrotherian species numbers and all of afrotherian disparity, and for each afrotherian taxon we included one or multiple non-afrotherian analogs. Regarding the stability of our results, we performed several jackknifing (resampling) analyses to test the robustness of the Procrustes distances and the PLS results, and these did not alter the outcome or the interpretation of our study. Following one of your remarks, we carried out the jackknifing on both the median and mean values, again showing strong agreement. We furthermore explained why phylogenetic and adaptive signals may overlap and that the former are not always indicative of neutral evolution in all clades. Finally, we uploaded all bony labyrinth surface models and the R code of our analyses, as you requested. In response to the other reviews, including the new one by Reviewer 4, we have made several further clarifications, placed our findings in a broader comparative context, and added a new distance-based metric that tests for convergent evolution while standardizing for phylogenetic relatedness.

Reviewer #3 (Remarks to the Author):

The revised manuscript by Grunstra and colleagues is substantially altered to improve clarity and reproducibility- including R scripts for future analyses. Detailed author responses satisfy my previous concerns. This current version will be a nice addition to the literature.

Response: Many thanks for your positive review and we are happy to read that we were able to assuage your previous concerns.

Reviewer #4 (Remarks to the Author):

Overall impressions

This study is an exploratory treatment of morphological convergence between afrotherian mammals and putative analogous mammals outside Afrotheria. The results begin to address one or two of the predictions of the hypothesis of high evolvability in the mammal ear (Le Maître et al., 2020). In general, the present manuscript is written clearly. The statistical techniques are appropriate to the questions and are approached carefully. The relevance of the findings to mammal evolution, to inner ear function, and to the debate about evolvability is discussed thoroughly. I primarily have questions for the authors (including on aspects of the statistical techniques), suggestions for additional citations, and thoughts about ear development.

The noteworthiness of this study relates to both methods and results. The authors use statistical measures that are otherwise commonplace pieces of many comparative works based on geometric morphometrics (distance and two-block partial least squares) but link them together with a set of contextual, ecological variables to explore the presence and strength of homoplastic patterns. The authors conclude that, indeed, there is morphological convergence in the inner ear of afrotherian and other mammals, and that the axes on which shape variation primarily falls are closely tied to ecology. The total workflow is unique, though I think the manuscript should cite earlier examples of studies that used morphological distances to assess convergence.

Overall, this manuscript is thought-provoking and well written. More pointed thoughts follow, as do line-by-line comments and references. Please note that some of my thoughts from my line-by-line comments are summarized in a generalized fashion in the sections that immediately follow this introduction.

Response: Thank you for your thorough review and your helpful feedback. We respond in a point-by-point response below. All edited text is highlighted in yellow in the revised manuscript.

Sample size

As other reviewers noted and the authors discussed in their rebuttal, the study's sample size is modest, relative to recent studies in Nature Communications that asked broadly similar questions, used broadly similar methods, and/or worked in the same organ system (e.g., Arbour et al., 2019; Singh et al., 2021; Mennecart et al., 2022). The authors defend the size of their sample, suggesting that any signals of convergence might be weakened if additional species were included. I looked for other studies that made similar arguments, but most references I could find relate to the effects of morphological noise on reconstructing phylogenies (in which larger taxonomic samples are often desired) (e.g., Gaubert et al., 2005; Revell et al., 2008), rather than exploring convergence. Wouldn't it be possible to expand the sample and still detect convergence if using a tanglegram (e.g., Zelditch et al., 2017; Stange, et al., 2018) and C-measures (Stayton, 2015)?

That all said, although the authors do not lay out a detailed history of why the taxa in their sample are analogs, the authors state upfront that there are a priori reasons to view afrotherians as being convergent with other mammals. I note that the authors did not set out with the goal to explicitly test whether convergence in afrotherians and other mammals happens more frequently than expected by chance. Answering that question presumably would require a larger sample and different methods (mentioned above and discussed below). But given the a priori presumption of convergence, this sample, which by all appearances adequately addresses the breadth of morphological and ecological disparity in afrotherians, seems to be of adequate size to answer the questions posed by the authors in the present study.

Response: Thank you, for your reassuring comments. Indeed, while estimates of the frequency of convergence would require larger samples, our approach is based on a priori assumptions about convergent pairs of species based on functional, ecological, and morphological data (see Table 1, column “basis for analogy”, fig. 1, and L. 90-100 and refs. therein). For this reason, we regard careful sample composition more important than sheer sample size. The C-measures by Stayton 2015 (and referenced in Zelditch et al. 2017) also require the prior identification of convergent species pairs, but this statistic additionally relies on estimates of ancestral states at multiple nodes (or some fossil evidence). We find the latter too unreliable in the case of convergent adaptations, especially in light of the underlying Brownian Motion model of neutral evolution and the limited number of species in our sample. A tanglegram would not require such a pairing and provide a visual impression of convergence (but see the criticism by de Vienne 2018 *Mol Biol Evol*).

Methods

The authors take a multivariate exploratory approach to quantify convergence between afrotherians and other mammals. The primary components of this approach are, first, a single summary statistic (a ratio of average Procrustes distances) calculated from a geometric morphometric analysis of shape, and second, an exploration of shape space (a two-block partial least squares regression analysis) with respect to contextual, ecological variables. The authors test the robustness of their findings with a jackknife analysis (i.e., removing different combinations of species to assess stability of the results). The logical flow of the analytical techniques is straightforward and the results relatively easy to interpret.

That said, the process of using average Procrustes distances to assess convergence resembles the Wheatsheaf index (Arbuckle et al., 2014), which tests for convergence using a ratio of Euclidean distances that accounts for phylogeny. Wheatsheaf was based on comparisons of Euclidean distances that were used in a morphometric context like the present study (Winemiller, 1991; Muschick et al., 2012). I am surprised that one or more such studies are not referred to here. Granted, the distances employed in the present study are Procrustes, so that is certainly an innovation relative to those other comparisons (which use Euclidean distances). However, the authors should cite one or more studies that used ratios of distances to assess morphometric convergence, as they are forerunners of the approach the authors use. Since the Procrustes distance is a number that could presumably be plugged into the Wheatsheaf formula, would the authors consider calculating a formal “Wheatsheaf” index (i.e., incorporating phylogeny) based on Procrustes distances for this sample?

I also was curious why C-measures were not deployed in this study to assess convergence, since I presume the ratio of Procrustes distances, like the Wheatsheaf index, might capture stasis and distinctiveness (Stayton, 2015). C-measures can quantify strength of convergence, which could be compared across pairs of analogues. The C-measures also can be a component of tests for whether convergence happens more often than expected by chance (as in Zelditch et al., 2017). Were C-measures not used because there was not an explicit test whether convergence in inner-ear form is more frequent than is expected by chance in mammals?

Response: Thank you for bringing these papers to our attention, and we fully agree that they are relevant here and deserve to be cited. Our own previous literature search did not yield these results and we came up with our distance-based approach independently. We are happy to cite these references here (see L. 119, 122, 151, 419). In fact, the Wheatsheaf index is basically the same as our ratio of average Procrustes distances with an additional step to correct for phylogenetic

relatedness (Procrustes distances are usually calculated as Euclidean distances between vectors of shape coordinates).

We have calculated the Wheatsheaf index (as per Arbuckle et al. 2014) for our data and included the results in the current revision. Very close to the ratio of average Procrustes distances (1.173), we obtained an index, w , of 1.183, demonstrating a shorter average distance (and thus stronger phenotypic similarity) between afrotherians and their analogues (“focal species”) compared to all other possible species pairs. We have incorporated this new analysis in lines 149-152 and 415-419 in our revision.

Figures and Tables

Generally the figures and tables are appropriate and support the text. I encourage the authors to update the color schemes of their figures for accessibility (i.e., for readers with vision disabilities). For figures 3 and 4, the red and green colors look identical under color-deficient proof settings. More specifically, the 3-D reconstructions of vestibular and cochlear shapes look identical in color, and the arboreal and subterranean datapoints look identical in color. I will leave any comments on individual figures and tables in a separate section below.

Response: You are right and we think accessibility is important. However, the colours in these figures are not essential as the species are all labelled separately and the most important grouping, those into afrotherians and non-afrotherians, is indicated by symbol type/shape, not colour. Also in the ear shape reconstructions, the different parts are unconnected and clearly distinguishable.

Line-by-line comments

Lines 47-48: I’m not sure what this statement adds. Is the implication that the pinna and ear canal vary by species more than the soft tissues of external ears of non-mammalian tetrapods do? If so, that point is not clear. Or is the point that only mammals evolved a pinna and ear canal? That may be so, but other groups of tetrapods have evolved complex specializations of the outer ear (or non-ear specializations that aid hearing). Birds have no pinna made of skin and connective tissue, but owls and many diurnal raptors have the facial disc (Knudsen & Konishi, 1979; von Campenhausen & Wagner, 2006), which varies among species, is deformable by musculature, and aids sound collection. Among owls, the skin of the outer ear (as well as the bone, in some cases) also varies by species and can aid sound collection due to asymmetry. Birds also have specialized feathers covering the ear that don’t impede sound conduction (Rival, 2005) and presumably serve a protective function. In crocodylians, the soft tissues surrounding the outer ear develop into a muscular set of scaly flaps that close to protect the deeper structures during submersion (“ear flaps”, Shute & Bellairs, 1955).

Response: Fair point, we have removed the mention of the mammalian pinna. We have rewritten the sentence to highlight the fact that only mammals have co-opted the angular bone of the jaw into the outer ear as a specialised bone to support the eardrum (L. 50-51). In other groups, the support of the eardrum is achieved either by another single bone not derived from mandibular structures, or by a combination of several bones, to which the angular bone can be included, but never alone.

Lines 50-57: How would the increase in overall developmental complexity of the middle ear impact the vestibular system? This topic is glossed over here, and the citations do not discuss mechanisms that would translate that increase in complexity to the vestibular system. It makes sense that increased developmental complexity of the middle ear (from an extra pharyngeal arch) could

release constraints on the middle ear and allow for more morphological variation in that system. Given that the middle ear ossicles are directly linked with the cochlea, it also is plausible that the auditory portion of the inner ear could be impacted by the additional evolutionary “knobs and dials”. Still, the inner ear derives from a separate embryonic structure. Why would we expect middle-ear changes to impact the semicircular canals, which are not auditory organs?

Response: A direct contribution to the vestibular system is indeed unlikely, but additional developmental factors acting on the middle ear contribute to the developmental independence (modularity) of inner and middle ears and thus also increases the evolutionary potential of the vestibular system. Also, as the cochlear and vestibular part of the inner ear are not completely independent morphologically and developmentally, changes that affect the cochlear part of the inner ear due to some evolution of the auditory system as a whole (i.e., the outer, middle and/or inner ear) may also affect the vestibular part. We extended the text to clarify this (L. 56-63).

Line 75: “comprising only few species”
I’d encourage the authors to write the approximate count.

Response: We added the number in the Introduction (L. 81) and again in the Methods (L. 321-322).

Lines 150-170: Several sentences in this section are grammatically a little hard to follow because of overuse of commas. I know there are space limitations, but I recommend breaking these results into smaller sentences for ease of reading.

Response: We rephrased and broke up the sentences for easier reading (L. 161-182).

Line 163: I suggest removing “subterraneous” (or at least changing to “subterranean”, which is used everywhere else in the manuscript). I think “fossorial” alone gets the point across, given that Table 2 states that subterranean habits are captured by the “fossoriality” locomotor mode.

Response: We replaced the occurrences of “subterraneous” by “subterranean” throughout. However, we kept the distinction between “subterranean” and “fossorial”, because they are not defined in the same way in the manuscript: fossorial is broader in that it means burrowing, but subterranean refers to underground-living (e.g., the hedgehog is fossorial but not subterranean).

Line 165: What does “curvature” mean in this context? The narrowness or tightness of the coil, or something else? It’s not clear.

Response: In this context, “curvature” referred to the tightness of the coil. We modified it for clarity (see L. 177).

Line 183: “too” is unnecessary because there is an “also” earlier in the sentence.

Response: Removed.

Line 191: “even” is unnecessary to understand the sentence.

Response: Removed.

Lines 209-210: I suggest writing “Convergent adaptations” (plural rather than singular) or explicitly using “homoplasy” or “pattern” at the beginning of the sentence. Some readers might think it’s circular to say that convergent adaptation (singular) is the most convincing evidence of adaptive evolution because “adaptation” could refer to the process or the resulting homoplastic pattern.

Response: We agree with the importance of distinguishing between pattern and process for convergent adaptation. We wrote “convergent adaptations” in plural and clarified that we study both the patterns and the process of evolutionary convergence (L. 121-126, 221, 223, 404-405, 423-424, 446-447).

Lines 213-214: Consider removing “were able to” and just writing “We identified”. It’s simpler and saves a few words.

Response: Done (L. 225).

Lines 216-220: “particularly” doesn’t seem applicable in the framing of the updated sentence. The sentence reads like the size of aquatic animals’ (as a group) canals is relative to others (as a group), so they would just be small, not particularly so. All else being equal, relatively smaller canals will be less sensitive, so the subsequent discussion about being aquatic would still be supported without the emphasis. Also, the sentence's readability would benefit from being broken into two or three sentences.

Response: We have broken up the sentence for better readability, and rephrased the first part to be more specific (L. 228-232).

Lines 404-410: The method (including permutation tests) deployed here sounds very similar to the Wheatsheaf index, minus phylogeny. The only difference I can see is that the distance used is Procrustes rather than Euclidean. If there is a difference that I have missed, could the authors please explain? I’ve read through Arbuckle et al. (2014), and the explanation of the Wheatsheaf workflow sounds just like the present study, just with a different kind of distance. Also, the C-measures (Stayton, 2015) are amenable to geometric morphometric data, as in Zelditch et al. (2017) and Stange et al. (2018). C-measures do not make assumptions about evolutionary process.

Response: Yes, you are right and we implemented the Wheatsheaf index in the revised manuscript (L. 149-152 and 415-419). Note, however, that, in addition to correcting for phylogeny, the Wheatsheaf index differs from our own approach in that the former constitutes a single ratio between the average distance among ‘focal’ species pairs (corresponding to our analogue pairs) and *all possible species pairs* (‘non-focal’ pairs), whereas in our approach we compared distances specifically between 3 types of pairs: afrotheria-analogue pairs, afrotheria-afrotheria pairs, and afrotheria-non-analogue pairs (taken from the non-afrotherian sample). Please see our response above for the C-measures.

Figures

Figure 1: It looks like the whiskers from the silhouette of *Sorex araneus* are duplicated and overlap the taxon name (only at its appearance near the bottom of the image).

Response: Good catch. We have corrected the figure.

Figure 2: It is a little challenging to see the landmarks in part A. The landmarks overlap with each other quite a lot because of the three-dimensionality of the structure. I don't know if there is an easy fix for that problem.

Response: Unfortunately it is not easy to distinguish most landmarks when the labyrinth is seen from a superior view; but we checked that all of these overlapping landmarks are easily distinguishable from the lateral view. We choose to show both views because these are the standard views when visualizing the bony labyrinth.

Figures 3 and 4: These are very nice images. Apart from my suggestion to update the color palette, these figures communicate the results well.

Response: Thank you.

Tables

Table 1: Out of curiosity, why are rodents like the kangaroo mouse not “microcursorial”? Are they too specialized for leaping/bounding?

Response: Correct. Rodents like kangaroo mice and jerboas engage in saltatorial (leaping) locomotion, whereas certain macroscelidean taxa are true “microcursors” (small-bodied fast runners; see also Lovegrove & Mowoe 2014 *J. Exp. Biol.* 217, 1316-25). This is also supported by the similarity in metatarsal-to-femur ratios of both fore- and hindlimbs in (micro)cursorial mammals. In saltatorial rodents, the hindlimbs are morphologically distinct from the forelimbs.

Table 2: Why include a “fully orthograde” state if none of the sampled taxa are fully orthograde?

Response: The current Afrotheria study is situated in a larger macroevolutionary research project by the authors on functional morphology of the inner ear, which includes a large variety of taxa, including humans, which are fully orthograde.

Reviewers' Comments:

Reviewer #4:

Remarks to the Author:

Overall impressions

I appreciate that the authors addressed each of my comments. I recommend the manuscript be accepted, though I ask that the authors consider the points I bring up below, particularly the reiteration of the point about accessibility.

Convergence and C-measure Discussion

"Response: Thank you, for your reassuring comments. Indeed, while estimates of the frequency of convergence would require larger samples, our approach is based on a priori assumptions about convergent pairs of species based on functional, ecological, and morphological data (see Table 1, column 'basis for analogy', fig. 1, and L. 90-100 and refs. therein). For this reason, we regard careful sample composition more important than sheer sample size. The C-measures by Stayton 2015 (and referenced in Zelditch et al. 2017) also require the prior identification of convergent species pairs, but this statistic additionally relies on estimates of ancestral states at multiple nodes (or some fossil evidence). We find the latter too unreliable in the case of convergent adaptations, especially in light of the underlying Brownian Motion model of neutral evolution and the limited number of species in our sample. A tanglegram would not require such a pairing and provide a visual impression of convergence (but see the criticism by de Vienne 2018 Mol Biol Evol)."

I'm not sure what the authors mean that they "find [ancestral state reconstruction] too unreliable in the case of convergent adaptations." Does the statement mean that such methods tend to reconstruct ancestors as having "average" morphology under a neutral model? If so, there are certainly other approaches to state reconstruction, including models that incorporate optima (see Webster & Purvis, 2002; Joy et al., 2016; e.g., Elliot & Mooers, 2014). The authors could even choose character states from the modern sample to represent maximally distinct ancestral states in calculation of the C-measures, with obvious caveats (Stayton, 2015). None of these methods are without drawbacks, but they could provide information beyond the Wheatsheaf+PLS approach. Even a tanglegram (if not ideal for visually discriminating among trees, per the de Vienne criticism) could provide a visual summary of the pattern of convergence.

Bottom line, I don't find the argument against using C-measures (or a different, model-based approach) particularly persuasive, **but given the sampling strategy I think the authors adequately addressed my initial concern on this topic.**

On pattern vs. process

"We agree with the importance of distinguishing between pattern and process for convergent adaptation. We wrote "convergent adaptations" in plural and clarified that we study both the patterns and the process of evolutionary convergence (L. 121-126, 221, 223, 404-405, 423-424, 446-447)."

The modifications are helpful and the additional discussion appreciated. I recommend the authors look at Mahler et al. (2017), however, for more context on pattern and process in studies of convergence. I think Mahler's Table 1 potentially provides additional (generalized) testable hypotheses for the overall ear-evolvability hypothesis. On the other hand, because the present study does not incorporate fossil evidence (nor ancestral reconstruction), whether some of the similarity is due to stasis (i.e., rather than convergence; Mahler et al., 2017, p. S21) was not tested. I suggest that the authors point out the caveat, even if it is contextualized with the adaptive signal that the authors recovered.

Figures and Tables

Original comment: ...I encourage the authors to update the color schemes of their figures for accessibility (i.e., for readers with vision disabilities). For figures 3 and 4, the red and green colors look identical under color-deficient proof settings. More specifically, the 3-D reconstructions of vestibular and cochlear shapes look identical in color, and the arboreal and subterranean datapoints look identical in color...

"Response: You are right and we think accessibility is important. However, the colours in these figures are not essential as the species are all labelled separately and the most important grouping, those into afrotherians and non-afrotherians, is indicated by symbol type/shape, not colour. Also in the ear shape reconstructions, the different parts are unconnected and clearly distinguishable."

I am puzzled why, if the authors agree, they took no action. If accessibility is important and there is information intended to be conveyed with the colors, then I reemphasize my initial comment and urge the authors to make the modification. After all, the authors argued in the previous round to keep the qualitative ecotypes in the figure, and the colors are mentioned in the figure caption. Otherwise, if the colors are not essential, why aren't shades of gray used instead?

The ear parts are identified by color in the figure 3 caption but unlabeled in the figure. The shape reconstructions look different from Figure 2 because of the transparency issues I mentioned in my first review. Perhaps most readers will be acquainted with labyrinth morphology enough to recognize what the simplified shapes represent, but if the aim is to appeal to a broader audience, then an accessible color scheme and/or labels (that parallel Fig. 2) on Figure 3 would be helpful. There is room in the white space of Figure 3 for labels.

One 5-color scheme that seems to work is black #000000, red #FF0000, magenta #FF00FF, cyan #00FFFF (with black outline because cyan can appear very light), and gold #FFD700.

Line-by-line comments

Line 108: “assumed”

I recommend “hypothesized”, rather than “assumed”, because this study is billed as an effort at testing the hypothesis from the 2020 paper.

Line 119: missing an end-parenthesis after “sensu53,54”

Literature Cited

Elliot, M.G., Mooers, A.Ø. Inferring ancestral states without assuming neutrality or gradualism using a stable model of continuous character evolution. *BMC Evol Biol* 14, 226 (2014).

<https://doi.org/10.1186/s12862-014-0226-8>

Joy JB, Liang RH, McCloskey RM, Nguyen T, Poon AFY (2016) Ancestral Reconstruction. *PLoS Comput Biol* 12(7): e1004763. <https://doi.org/10.1371/journal.pcbi.1004763>

Mahler, D. Luke, Marjorie G. Weber, Catherine E. Wagner, and Travis Ingram. Pattern and Process in the Comparative Study of Convergent Evolution. *The American Naturalist* 2017 190:S1, S13-S28.

Webster, Andrea J. and Purvis, Andy. 2002. Testing the accuracy of methods for reconstructing ancestral states of continuous characters. *Proc. R. Soc. Lond. B.* 269:143–149.

<http://doi.org/10.1098/rspb.2001.1873>

REVIEWER'S COMMENTS

Reviewer #4 (Remarks to the Author):

Overall impressions

I appreciate that the authors addressed each of my comments. I recommend the manuscript be accepted, though I ask that the authors consider the points I bring up below, particularly the reiteration of the point about accessibility.

Response: we thank the reviewer for their positive evaluation and recommendation for publication. We address their remaining comments individually below.

Convergence and C-measure Discussion

"Response: Thank you, for your reassuring comments. Indeed, while estimates of the frequency of convergence would require larger samples, our approach is based on a priori assumptions about convergent pairs of species based on functional, ecological, and morphological data (see Table 1, column 'basis for analogy', fig. 1, and L. 90-100 and refs. therein). For this reason, we regard careful sample composition more important than sheer sample size. The C-measures by Stayton 2015 (and referenced in Zelditch et al. 2017) also require the prior identification of convergent species pairs, but this statistic additionally relies on estimates of ancestral states at multiple nodes (or some fossil evidence). We find the latter too unreliable in the case of convergent adaptations, especially in light of the underlying Brownian Motion model of neutral evolution and the limited number of species in our sample. A tanglegram would not require such a pairing and provide a visual impression of convergence (but see the criticism by de Vienne 2018 Mol Biol Evol)."

I'm not sure what the authors mean that they "find [ancestral state reconstruction] too unreliable in the case of convergent adaptations." Does the statement mean that such methods tend to reconstruct ancestors as having "average" morphology under a neutral model? If so, there are certainly other approaches to state reconstruction, including models that incorporate optima (see Webster & Purvis, 2002; Joy et al., 2016; e.g., Elliot & Mooers, 2014). The authors could even choose character states from the modern sample to represent maximally distinct ancestral states in calculation of the C-measures, with obvious caveats (Stayton, 2015). None of these methods are without drawbacks, but they could provide information beyond the Wheatsheaf+PLS approach. Even a tanglegram (if not ideal for visually discriminating among trees, per the de Vienne criticism) could provide a visual summary of the pattern of convergence.

Bottom line, I don't find the argument against using C-measures (or a different, model-based approach) particularly persuasive, but given the sampling strategy I think the authors adequately addressed my initial concern on this topic.

Response: we appreciate the reviewer's feedback and it is something we will keep in mind for future work. For this manuscript, however, we feel that by now we have offered adequate support for convergent evolution in our sample through various analyses and statistics.

On pattern vs. process

"We agree with the importance of distinguishing between pattern and process for convergent adaptation. We wrote "convergent adaptations" in plural and clarified that we study both the patterns and the process of evolutionary convergence (L. 121-126, 221, 223, 404-405, 423-424, 446-447)."

The modifications are helpful and the additional discussion appreciated. I recommend the authors look at Mahler et al. (2017), however, for more context on pattern and process in studies of convergence. I think Mahler's Table 1 potentially provides additional (generalized) testable hypotheses for the overall ear-evolvability hypothesis. On the other hand, because the present study does not incorporate fossil evidence (nor ancestral reconstruction), whether some of the similarity is due to stasis (i.e., rather than convergence; Mahler et al., 2017, p. S21) was not tested. I suggest that the authors point out the caveat, even if it is contextualized with the adaptive signal that the authors recovered.

Response: We thank the reviewer for helping us think about further ways to test our ear evolvability hypothesis, and the Mahler et al. 2017 reference is helpful here. We have added a short note on the lack of fossil evidence in the manuscript (L. 272-277). Nonetheless, evolutionary stasis is unlikely to explain the signal we interpret as convergent evolution, because we are dealing with up to 250 million years of separate evolution since the last common ancestor between afrotherians and their analogues. Furthermore, stasis would imply, given that 36 out of 41 afrotheria-analogue pairs share the same last common ancestor, that all these afrotherian-analogue pairs retained the same ancestral state while only non-analogues changed their morphology, which is both unparsimonious and clearly inconsistent with the variation we observe. Moreover, it is well-established that sea cows and cetaceans independently acquired an obligatory aquatic lifestyle from a terrestrial ancestor.

Figures and Tables

Original comment: ...I encourage the authors to update the color schemes of their figures for accessibility (i.e., for readers with vision disabilities). For figures 3 and 4, the red and green colors look identical under color-deficient proof settings. More specifically, the 3-D reconstructions of vestibular and cochlear shapes look identical in color, and the arboreal and subterranean datapoints look identical in color...

"Response: You are right and we think accessibility is important. However, the colours in these figures are not essential as the species are all labelled separately and the most important grouping, those into afrotherians and non-afrotherians, is indicated by symbol type/shape, not colour. Also in the ear shape reconstructions, the different parts are unconnected and clearly distinguishable."

I am puzzled why, if the authors agree, they took no action. If accessibility is important and there is information intended to be conveyed with the colors, then I reemphasize my initial comment and urge the authors to make the modification. After all, the authors argued in the previous round to keep the qualitative ecotypes in the figure, and the colors are mentioned in the figure caption. Otherwise, if the colors are not essential, why aren't shades of gray used instead?

The ear parts are identified by color in the figure 3 caption but unlabeled in the figure. The shape reconstructions look different from Figure 2 because of the transparency issues I mentioned in my first review. Perhaps most readers will be acquainted with labyrinth morphology enough to recognize what the simplified shapes represent, but if the aim is to appeal to a broader audience, then an accessible color scheme and/or labels (that parallel Fig. 2) on Figure 3 would be helpful. There is room in the white space of Figure 3 for labels.

One 5-color scheme that seems to work is black #000000, red #FF0000, magenta #FF00FF, cyan #00FFFF (with black outline because cyan can appear very light), and gold #FFD700.

Response: We agree and have now adapted Figs 3 and 4 following the recommendations of the reviewer. We added labels that identify the different ear parts on the shape patterns (with corresponding definitions in the captions), which we think are even more informative than colors, and we have used the suggested color scheme for the PLS plots (Figs. 3c, 3f, 4c and 4f), which we tested for visual accessibility.

Line-by-line comments

Line 108: “assumed”

I recommend “hypothesized”, rather than “assumed”, because this study is billed as an effort at testing the hypothesis from the 2020 paper.

Response: modified

Line 119: missing an end-parenthesis after “sensu53,54”

Response: modified

Reviewer #4 (Remarks on code availability):

Yes, I was able to install and run the code. In general, the code was annotated well and there is a README file with sufficient instruction to install it and run. I was able to reproduce the results and figures using the code.

Response: great.